# Order Matters in Retrosynthesis: Structure-aware Generation via Reaction-Center-Guided Discrete Flow Matching

**Chenguang Wang** [*1 2]   **Zihan Zhou** [*1 2]   **Lei Bai** [2]   **Tianshu Yu** [1 2]

## Abstract

Template-free retrosynthesis methods treat the task as black-box sequence generation, limiting learning efficiency, while semi-template approaches rely on rigid reaction libraries that constrain generalization. We address this gap with a key insight: atom ordering in neural representations matters. Building on this insight, we propose a structure-aware template-free framework that encodes the two-stage nature of chemical reactions as a positional inductive bias. By placing reaction center atoms at the sequence head, our method transforms implicit chemical knowledge into explicit positional patterns that the model can readily capture. The proposed RetroDiT backbone, a graph transformer with rotary position embeddings, exploits this ordering to prioritize chemically critical regions. Combined with discrete flow matching, our approach decouples training from sampling and enables generation in 20–50 steps versus 500 for prior diffusion methods. Our method achieves state-of-the-art performance on both USPTO-50k (61.2% top-1) and the large-scale USPTO-Full (51.3% top-1) with predicted reaction centers. With oracle centers, performance reaches 71.1% and 63.4% respectively, surpassing foundation models trained on 10 billion reactions while using orders of magnitude less data. Ablation studies further reveal that structural priors outperform brute-force scaling: a 280K-parameter model with proper ordering matches a 65M-parameter model without it.

## 1. Introduction

Retrosynthesis, the task of identifying precursor molecules to synthesize a target compound, is fundamental to organic chemistry and drug discovery (Vleduts, 1963; Corey & Wipke, 1969). Single-step retrosynthesis prediction, which determines the immediate reactants for a given product, underpins multi-step route planning and has driven extensive research in machine learning (Coley et al., 2019; Schwaller et al., 2020; Chen et al., 2020; Xie et al., 2022; Tripp et al., 2022; Liu et al., 2023; Tripp et al., 2023; Zhong et al., 2023b).

Existing approaches to single-step retrosynthesis fall into three paradigms. Template-based methods (Coley et al., 2017; Segler & Waller, 2017; Dai et al., 2019; Chen & Jung, 2021; Yan et al., 2022) match products against a library of reaction templates extracted from known reactions. While interpretable, they generalize poorly to novel reactions not covered by the template library. Semi-template methods (Shi et al., 2020; Yan et al., 2020; Wang et al., 2021; Chen et al., 2023; Gao et al., 2022; Zhong et al., 2023a) decompose the task into reaction center identification followed by synthon completion, making the learning problem more tractable. However, they still rely on templates or predefined rules for completing synthons, limiting generation flexibility. Template-free methods (Zheng et al., 2019; Sacha et al., 2021; Tu & Coley, 2022; Zeng et al., 2024; Igashov et al., 2023; Yao et al., 2024; Zhong et al., 2022; Han et al., 2024; Yadav et al., 2025) directly generate reactants from products in an end-to-end manner, offering maximum flexibility. Yet by treating the task as a black-box transformation, they ignore the intrinsic structure of chemical reactions, leading to inefficient learning.

These observations raise a natural question: *Can we combine the structural insights of semi-template methods with the generation flexibility of template-free approaches?*

In this work, we address this question with a key insight: single-step retrosynthesis naturally decomposes into two stages, identifying the reaction center and generating the corresponding reactant structures. This two-stage structure explains why template-free methods struggle with learning efficiency. They must implicitly learn both tasks simultane-

[1]School of Data Science, The Chinese University of Hong Kong, Shenzhen [2]Shanghai Artificial Intelligence Laboratory. Correspondence to: Tianshu Yu <yutianshu@cuhk.edu.cn>.

*Proceedings of the 43rd International Conference on Machine Learning*, Seoul, South Korea. PMLR 306, 2026. Copyright 2026 by the author(s).

ously, without guidance on the problem's inherent decomposition.

Building on this insight, we propose a structure-aware template-free framework that combines structural decomposition with generation flexibility. The core idea is reaction-center-aware atom ordering: by designating reaction center atoms as root nodes and traversing the molecular graph accordingly, we place reaction centers at the beginning of the node sequence. This encodes the two-stage structure as a positional inductive bias. To capture this ordering, we introduce RetroDiT, a graph transformer backbone equipped with Rotary Position Embeddings (RoPE) (Su et al., 2024). Built upon Discrete Flow Matching (Campbell et al., 2024; Gat et al., 2024; Qin et al., 2024), our framework decouples training from sampling, enabling generation in 20–50 steps versus 500 for prior diffusion methods.

Our framework adopts a modular design. During training, ground-truth reaction centers from atom mapping serve as root nodes; during inference, a separately trained predictor provides the roots. This separation allows each component to be optimized independently, and future improvements in reaction center prediction will directly boost overall performance.

We evaluate our approach on USPTO-50k and USPTO-Full (Lowe, 2012; Zhong et al., 2023b). With oracle reaction centers, our method achieves 71.1% top-1 accuracy on USPTO-50k and 63.4% on USPTO-Full, establishing strong upper bounds that surpass foundation models trained on 10 billion samples. With predicted reaction centers, we achieve 61.2% and 51.3% respectively, outperforming existing template-free methods by over 10 points on USPTO-50k. Training converges 6× faster than previous baselines.

Ablation studies reveal two key findings. First, positional inductive bias is more effective than scaling model size or training data: a 280K-parameter model with proper ordering matches a 65M-parameter model without it. Second, reaction center prediction is the primary performance bottleneck, providing clear direction for future work.

Our main contributions are:

- We introduce a structure-aware template-free paradigm that encodes the two-stage nature of retrosynthesis as a positional inductive bias through atom ordering, eliminating the need for explicit templates;

- We propose RetroDiT, a graph backbone with rotary position embeddings to capture structural priors in ordered molecular graphs, and achieve state-of-the-art performance with 6× training speedup and up to 25× fewer sampling steps via discrete flow matching;

- We provide systematic analysis demonstrating that structure-aware inductive bias outperforms brute-force

scaling, identifying reaction center prediction as the key bottleneck for future improvement.

## 2. Related Work

**Single-step Retrosynthesis Prediction.** Existing approaches fall into three paradigms. Template-based methods (Coley et al., 2017; Segler & Waller, 2017; Dai et al., 2019; Chen & Jung, 2021; Yan et al., 2022; Gaiński et al., 2025) match targets against predefined reaction libraries, providing interpretability at the cost of limited generalization. Semi-template methods (Shi et al., 2020; Yan et al., 2020; Wang et al., 2021; Chen et al., 2023; Gao et al., 2022; Zhong et al., 2023a) decompose the task into reaction center identification and synthon completion, reducing learning difficulty but relying on rigid completion rules. Template-free methods (Zheng et al., 2019; Sacha et al., 2021; Tu & Coley, 2022; Zhong et al., 2022; Yao et al., 2024; Han et al., 2024; Zeng et al., 2024) formulate retrosynthesis as end-to-end generation using sequence Transformers or graph generative models (Igashov et al., 2023; Yadav et al., 2025). Recent work further scales this approach through large language models pretrained on massive reaction corpora (Deng et al., 2025; Wang et al., 2025; Zhao et al., 2025b; Ozer et al., 2025; Zhang et al., 2025). Our method belongs to the template-free paradigm but incorporates structural insights from semi-template approaches through positional inductive biases, without requiring templates or completion rules.

**Inductive Biases in Retrosynthesis.** Prior work has explored various inductive biases to simplify learning. Topological approaches exploit graph-level constraints, using dual representations to highlight reaction centers (Sun et al., 2025) or enforcing chemical plausibility through embedding constraints (Somnath et al., 2021; Zhao et al., 2025a). Sequence-level approaches focus on alignment strategies, from root-aligned SMILES (Zhong et al., 2022; Yao et al., 2024) to Maximum Common Substructure matching (Hu et al., 2025), to reduce the search space. Other work addresses prediction uncertainty through data augmentation (Zhong et al., 2022; Sun et al., 2025) or round-trip verification (Yadav et al., 2025). Our approach differs by encoding structural priors directly into the positional encoding scheme, allowing a standard Transformer to capture reaction-center-aware patterns without specialized architectures.

**Discrete Generative Models.** Graph generation has evolved from continuous diffusion (Song et al.) to discrete denoising frameworks (Austin et al., 2021; Vignac et al., 2022; Lou et al.; Sahoo et al., 2024). Flow Matching (Lipman et al., 2024) offers a simulation-free alternative with efficient training, and recent extensions to discrete spaces (Campbell et al., 2024; Gat et al., 2024) and graphs (Qin et al., 2024) have shown strong results. We build on Discrete Flow Match-

ing for its decoupled training and sampling, which enables efficient generation in 20–50 steps while maintaining high accuracy.

## 3. Background

**Graph Notation.** We represent a molecule as an undirected graph $G = (x^{1:N}, e^{1:i<j:N})$ with $N$ nodes. The node features $x^{1:N} = (x^{(n)})_{1 \leq n \leq N}$ denote atom attributes, and edge features $e^{1:i<j:N} = (e^{(i\bar{j})})_{1 \leq i < j \leq N}$ denote bond attributes.

**Single-step Retrosynthesis.** Given a product molecule graph $G_P$, the goal is to generate the corresponding reactant graph $G_R$ by modeling the conditional distribution $p(G_R|G_P)$. We assume nodes in $G_P$ and $G_R$ are index-aligned via atom mapping: the $n$-th node in $G_P$ maps to the $n$-th node in $G_R$.

**Reaction Center.** We define the reaction center $\mathcal{S}_{RC} \subseteq \{1, \ldots, N\}$ as the subset of atoms in $G_P$ involved in the chemical transformation. An atom $n$ belongs to $\mathcal{S}_{RC}$ if it satisfies either of the following conditions:

- *Topological change*: The atom is connected to a bond that is formed, broken, or changed in bond order.

- *Property change*: The atom's formal charge, hydrogen count, chirality, aromaticity, or hybridization differs between $G_P$ and $G_R$.

This definition captures a wider range of reaction types than prior work such as G2Gs (Shi et al., 2020), which focuses primarily on bond changes. We detail the eight specific reaction center categories and their extraction logic in Appendix A.

## 4. Method

### 4.1. Structure-Aware Graph Representation

Standard graph neural networks and transformers treat molecular graphs as permutation-invariant structures, where absolute node indices carry no semantic meaning. While suitable for predicting global molecular properties, this design overlooks a key aspect of retrosynthesis: the transformation is strictly local. Specific atoms at the reaction center drive the reaction, while the rest of the molecule serves as a scaffold. A permutation-invariant model must implicitly rediscover these active sites for every prediction, wasting learning capacity.

Our core idea is to encode chemical knowledge through node ordering. By imposing a consistent, structure-aware ordering, we transform the abstract task of "where does the reaction occur" into a concrete positional pattern that the model can directly learn.

**Reaction-Center-Rooted Atom Ordering.** For a product graph $G_P$ with reaction center atoms $\mathcal{S}_{RC}$, we generate the node sequence by rooting the graph traversal at a specific reaction center atom $a \in \mathcal{S}_{RC}$. This places reaction center atoms at the beginning of the sequence, with remaining atoms ordered by their topological distance from the root. Implementation details using RDKit canonicalization are provided in Appendix B.

Since a reaction may involve multiple reactive atoms, we apply data augmentation during training: for a product with $|\mathcal{S}_{RC}|$ reaction center atoms, we create a separate training sample rooted at each atom $a \in \mathcal{S}_{RC}$. This expands the training data and compels the model to learn reaction patterns from the perspective of every active site.

**Leaving Group Placeholders.** Reactants often contain atoms absent in the product, such as leaving groups. To accommodate these within a fixed-size generation framework, we append $K$ dummy nodes to the tail of the ordered sequence. The final node sequence becomes $V_{\text{final}} = [V_{\text{ordered}}, d_1, \ldots, d_K]$, where $V_{\text{ordered}}$ contains the product atoms and $K$ is a hyperparameter controlling the maximum number of leaving group atoms.

**The Head-Body-Tail Structure.** This ordering strategy yields a structured sequence format: the *head* contains reaction center atoms, the *body* encodes the molecular scaffold, and the *tail* provides placeholders for leaving groups. Chemical logic thus becomes positional logic. However, a standard graph encoder would still ignore this ordering due to permutation invariance. Capturing this positional inductive bias requires a backbone that can distinguish sequence positions, which we introduce next.

### 4.2. RetroDiT Architecture

To learn the time-dependent transition for discrete flow matching, we design RetroDiT, a graph transformer backbone tailored for ordered molecular graphs. We first provide an architectural overview, then explain how Rotary Position Embeddings capture the ordering-based inductive bias.

**Architectural Overview.** RetroDiT processes node and edge features through $L$ transformer layers. Each layer consists of three components: (1) multi-head self-attention over nodes, modulated by edge information; (2) edge updates conditioned on attention patterns; and (3) feed-forward networks for both node and edge features. We employ additive time conditioning. This design allows the model to adjust its behavior across the generation trajectory. Full architectural details are provided in Appendix D.

**Capturing Positional Bias via RoPE.** The efficacy of our structure-aware representation depends on the model's ability to distinguish sequence positions. Standard graph transformers use learnable positional encodings or topological features (Ying et al., 2021), which do not explicitly encode

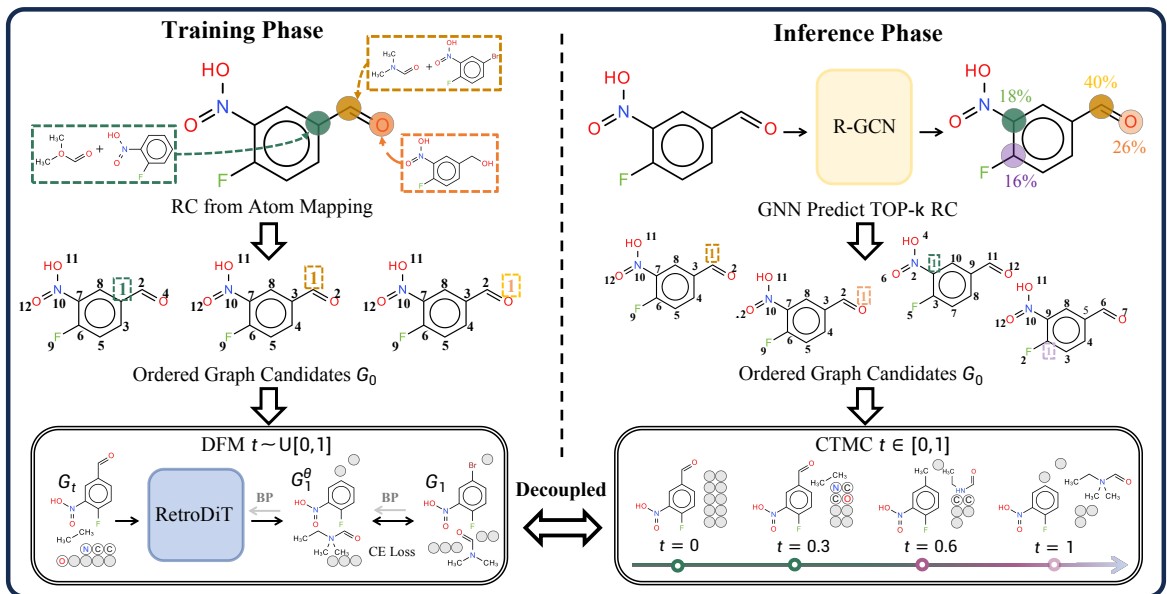

*Figure 1.* **Overview of the structure-aware retrosynthesis framework. Left (Training):** Ground-truth reaction centers are identified via atom mapping, generating multiple ordered graph candidates $G_0$ with different RC atoms as roots. Discrete flow matching trains RetroDiT to predict the target reactants $G_1$ from intermediate states $G_t$. **Right (Inference):** An R-GCN predicts reaction center probabilities, and top-$k$ candidates (4 in the illustration) are used to generate ordered graphs. CTMC sampling progressively transforms the product ($t = 0$) into reactants ($t = 1$). The training and inference pipelines are decoupled, allowing independent optimization of each component.

our reaction-center-rooted ordering. Instead, we adopt Rotary Position Embeddings (RoPE) (Su et al., 2024), which encode relative positions through rotation of query and key vectors:

$$\text{Attention}(q_i, k_j) = (R_i q_i)^\top (R_j k_j) = q_i^\top R_{j-i} k_j, \quad (1)$$

where $R$ is a rotation matrix and $j - i$ represents the relative position between nodes.

In our ordering scheme, relative position directly reflects topological distance from the reaction center. Nodes closer to the head (small indices) are at or near the reaction center; nodes at the tail (large indices) are dummy placeholders for leaving groups. RoPE allows attention weights to depend on this relative distance, enabling the model to learn position-dependent patterns, reducing the learning burden and improving sample efficiency.

Without positional encoding, a permutation-invariant model cannot distinguish dummy nodes from scaffold atoms, forcing it to learn this pairing implicitly. Detailed ablation results on positional encoding are demonstrated in Section 5.3.

### 4.3. Discrete Flow Matching for Retrosynthesis

We formulate retrosynthesis as a discrete flow matching problem, where the source distribution is the product graph and the target is the reactant graph. Unlike standard generative tasks that start from uniform noise, our flow interpolates between two structured molecular states, preserving chemi-

cal information throughout the generation process.

**Problem Setup.** Let $G_0 = G_P$ denote the product graph (source) and $G_1 = G_R$ denote the reactant graph (target). We construct a probability path that interpolates between these states over $t \in [0, 1]$:

$$p_{t|0,1}(G_t|G_0, G_1) = t \cdot \delta(G_t, G_1) + (1-t) \cdot \delta(G_t, G_0). \quad (2)$$

At intermediate times, the graph is either the product (with probability $1 - t$) or the reactant (with probability $t$). This linear interpolation provides a direct path between the two molecular structures.

**Training.** We train a neural network $f_\theta(G_t, t)$ to predict the target distribution $p_{1|t}^\theta(\cdot|G_t)$ given an intermediate graph $G_t$. Intuitively, the model learns to "see through" the noisy intermediate state and predict what the final reactant should be. The training objective is the cross-entropy loss:

$$\mathcal{L}_{\text{Retro}}(\theta) = \mathbb{E}_{t,(G_0,G_1),G_t} \left[ \text{CE}\left( G_1, p_{1|t}^\theta(\cdot|G_t) \right) \right], \quad (3)$$

where $t \sim \mathcal{U}(0, 1)$, $(G_0, G_1) \sim \mathcal{D}$, and $G_t \sim p_{t|0,1}$. This objective is simulation-free: we sample $(t, G_t)$ pairs directly without solving differential equations, enabling fast training. The complete training procedure is summarized in Algorithm 1.

**Inference.** At test time, we evolve the product graph from $t = 0$ to $t = 1$ by simulating a continuous-time Markov chain (CTMC). The key idea is to use the trained network's

**Algorithm 1** Structure-Aware DFM Training

**Require:** Dataset $\mathcal{D} = \{(G_P, G_R)\}$, dummy node count $K$, model $f_\theta$
1: **repeat**
2:     Sample batch $\{(G_P, G_R)\} \sim \mathcal{D}$
3:     $\mathcal{B}_{\text{aug}} \leftarrow \emptyset$
4:     **for** each $(G_P, G_R)$ in batch **do**
5:         Extract reaction centers $\mathcal{S}_{\text{RC}}$ from atom mapping
6:         **for** each atom $a \in \mathcal{S}_{\text{RC}}$ **do**
7:             $V_{\text{ordered}} \leftarrow \text{RootedOrdering}(G_P, a)$
8:             $G_0 \leftarrow \text{BuildSource}(G_P, V_{\text{ordered}}, K)$
9:             $G_1 \leftarrow \text{BuildTarget}(G_R, V_{\text{ordered}}, K)$
10:            Add $(G_0, G_1)$ to $\mathcal{B}_{\text{aug}}$
11:         **end for**
12:     **end for**
13:     **for** each $(G_0, G_1) \in \mathcal{B}_{\text{aug}}$ **do**
14:         Sample $t \sim \mathcal{U}(0, 1)$ and $G_t \sim p_{t|0,1}(G_t|G_0, G_1)$
15:         Predict $p_{1|t}^\theta \leftarrow f_\theta(G_t, t)$
16:         Compute $\mathcal{L} \leftarrow \text{CE}(G_1, p_{1|t}^\theta)$
17:         Update $\theta$ via gradient descent
18:     **end for**
19: **until** convergence

**Algorithm 2** Modular Inference Pipeline

1: **Input:** Product $G_P$, RC Predictor $\phi$, RetroDiT $f_\theta$, sampling steps $S$, repeat times $M$, $\mathcal{G}_{\text{reactants}} \leftarrow \emptyset$
2: Obtain top-k predicted RCs: $\mathcal{R} \leftarrow \phi(G_P)$
3: **for** $m = 1$ **to** $M$ **do**
4:     Sample root $r \in \mathcal{R}$
5:     $V_{\text{ordered}} \leftarrow \text{Canonicalize}(G_P, \text{root} = r)$
6:     $G_0 \leftarrow \text{Sort}(G_P, V_{\text{ordered}})$
7:     **for** $i = 0$ **to** $S - 1$ **do**
8:         $t \leftarrow i/S, \quad \Delta t \leftarrow 1/S$
9:         Predict posterior: $p_{1|t}^\theta \leftarrow f_\theta(G_t, t)$
10:         Compute rate matrix $R_t^\theta(G_t, \cdot|G_0)$ via Eq. 5
11:         Sample next state $G_{t+\Delta t}$ via Eq. 4, $G_t \leftarrow G_{t+\Delta t}$
12:     **end for**
13:     Add $G_{t=1}$ to $\mathcal{G}_{\text{reactants}}$
14: **end for**
15: **Return** $\mathcal{G}_{\text{reactants}}$

predictions to construct a rate matrix $R_t^\theta$ that governs transitions between graph states:

$$p^\theta(G_{t+\Delta t} = G'|G_t, G_0) \approx \delta_{G_t, G'} + R_t^\theta(G_t, G'|G_0)\Delta t. \tag{4}$$

Starting from $G_0 = G_P$, we iteratively sample the next state using this transition probability until reaching $t = 1$. The rate matrix $R_t^\theta$ is computed by taking an expectation over the network's predicted target distribution:

$$R_t^\theta(G_t, G'|G_0) = \mathbb{E}_{G_1 \sim p_{1|t}^\theta(\cdot|G_t)}\left[R_t(G_t, G'|G_0, G_1)\right], \tag{5}$$

where $R_t(G_t, G'|G_0, G_1)$ is the conditional rate matrix derived from the linear interpolation path. While the target-conditional rate $R_t(G_t, G'|G_0, G_1)$ can be constructed in various ways, we adopt the specific formulation from De-FoG (Qin et al., 2024) to enforce the linear interpolation path:

$$R_t(G_t, G'|G_0, G_1)$$
$$= \frac{\text{ReLU}(\partial_t p_{t|0,1}(G'|G_0, G_1) - \partial_t p_{t|0,1}(G_t|G_0, G_1))}{Z_t^{>0} p_{t|0,1}(G_t|G_0, G_1)}, \tag{6}$$

where $Z_t^{>0}$ is a normalization constant. In practice, we use Euler discretization with $S$ steps, where each step has size $\Delta t = 1/S$. As shown in Appendix G.3, our experiments show that $S = 20$–$50$ steps suffice for high-quality generation, compared to 500 steps required by prior diffusion-based methods (Igashov et al., 2023).

## 4.4. Modular Inference Pipeline

During training, ground-truth reaction centers from atom mapping provide the root nodes for ordering. At inference time, however, the reaction center is unknown. We address this with a two-stage pipeline that separates reaction center prediction from structure generation.

**Stage I: Reaction Center Prediction.** We train a lightweight Relational Graph Convolutional Network (R-GCN) (Schlichtkrull et al., 2018) to predict reaction centers. The model scores each atom in $G_P$ based on its likelihood of belonging to $\mathcal{S}_{\text{RC}}$, framing the task as binary classification. To handle the multi-modal nature of retrosynthesis, where multiple valid disconnection sites may exist, we retain the top-$k$ atoms as candidate roots. Predictor details are provided in Appendix E.2.

**Stage II: Structure-Aware Generation.** For each candidate root from Stage I, we construct the ordered sequence using reaction-center-rooted ordering (Section 4.1), then generate reactants using RetroDiT with CTMC sampling (Sections 4.2–4.3). Each candidate produces one reactant proposal; we collect all proposals and rank them by model likelihood.

**Benefits of Modularity.** This design offers two practical benefits. First, *efficiency*: by solving "where to react" separately, the generative model focuses on "how to react," enabling high-quality generation in few steps. Second, *upgradability*: the two components can be improved independently. Future advancements in either reaction center prediction accuracy or generative backbone architecture can be seamlessly integrated to boost overall performance without necessitating a full system retrain. Our analysis in Section 5.4 shows that reaction center prediction is currently the primary bottleneck, making this modularity particularly valuable.

The complete inference procedure is summarized in Algorithm 2, and Figure 1 provides an overview of the entire framework, illustrating how the training and inference pipelines share the same RC-rooted ordering scheme while remaining independently optimizable.

# 5. Experiments

## 5.1. Experimental Setup

**Datasets.** We evaluate our framework on two widely adopted benchmark datasets derived from the US Patent Office (USPTO) database. USPTO-50k contains approximately 50,000 atom-mapped reactions and serves as the standard retrosynthesis benchmark. USPTO-Full contains approximately 1 million reactions, providing a larger-scale testbed for evaluating scalability. We use the standard 80%–10%–10% split for both datasets (Maziarz et al., 2025).

**Baselines.** We compare against methods from three paradigms: (1) Template-based: RetroSim (Coley et al., 2017), NeuralSym (Segler & Waller, 2017), GLN (Dai et al., 2019), LocalRetro (Chen & Jung, 2021), and RetroComposer (Yan et al., 2022); (2) Semi-template: G2G (Shi et al., 2020), RetroXpert (Yan et al., 2020), Retro-Prime (Wang et al., 2021), G$^2$Retro (Chen et al., 2023), SemiRetro (Gao et al., 2022), and Graph2Edits (Zhong et al., 2023a); (3) Template-free: SCROP (Zheng et al., 2019), MEGAN (Sacha et al., 2021), Graph2SMILES (Tu & Coley, 2022), Retroformer (Wan et al., 2022), Retro-Bridge (Igashov et al., 2023), Ualign (Zeng et al., 2024), NAG2G (Yao et al., 2024), R-SMILES (Zhong et al., 2022), EditRetro (Han et al., 2024), and RetroFlow (Yadav et al., 2025). We also compare against RSGPT (Deng et al., 2025), a large language model pretrained on 10 billion reactions. While delivering superior performance, it relies on massive computational resources and a complex multi-stage pipeline (see Appendix E.1 for a detailed comparison).

**Metrics.** Following Maziarz et al. (2025), we report Top-$k$ Exact Match Accuracy ($k \in \{1, 3, 5, 10\}$), which measures whether the ground-truth reactants appear in the top-$k$ predictions. We also report Round-Trip Accuracy in Appendix F.1, which verifies that generated reactants can reproduce the target product when passed through a forward synthesis model, providing a measure of chemical validity. Detailed definitions are in Appendix E.3.

**Implementation.** Our main experiments use RetroDiT-Large (8M parameters) with 50 sampling steps. For comparison, prior diffusion-based methods such as RetroBridge require 500 steps. We use a lightweight GNN for reaction center prediction. Full training and sampling configurations are provided in Appendix E.

## 5.2. Main Results

**Performance on USPTO-50K.** Table 1 presents the comparative results on the USPTO-50k benchmark with reaction classes unknown. Our framework, utilizing a lightweight GNN-based reaction center predictor, achieves a Top-1 accuracy of **61.2%**, demonstrating remarkable performance against established baselines.

Crucially, our method significantly outperforms Retro-Bridge (Igashov et al., 2023) and RetroSynFlow (Yadav et al., 2025), the representative diffusion-based and flow-based generative baselines, by margins of 10.4% (61.2% vs. 50.8%) and 1.2% (61.2% vs. 60.0%) in Top-1 accuracy, respectively. This substantial gap validates our core claim: incorporating structural inductive bias via atom ordering is far more effective than treating graph generation as a black-box diffusion process. Furthermore, our method outperforms all semi-template methods, including Graph2Edits (55.1%) and G2Retro (54.1%), indicating that our "structure-aware template-free" paradigm successfully combines the precision of decomposition with the flexibility of generative modeling.

Compared to RSGPT (Deng et al., 2025), a Large Language Model pretrained on 10 billion reactions, our method achieves highly competitive results (61.2% vs. 63.4%) using *orders of magnitude less data* (only 50k training samples) and significantly fewer parameters. This suggests that the positional inductive bias serves as a data-efficient alternative to brute-force scaling, allowing the model to capture chemical rules without massive pretraining.

**Scalability on USPTO-Full.** To evaluate scalability, we report results on the larger USPTO-Full dataset in Table 2. Our method achieves 51.3% Top-1 accuracy with predicted reaction centers, surpassing all existing template-based, semi-template, and non-LLM template-free baselines (e.g., NAG2G at 49.7% and R-SMILES at 48.9%). Moreover, under the Oracle setting, our generative backbone achieves 63.4% Top-1 accuracy, notably outperforming RSGPT (59.2%). This result is particularly significant as it demonstrates that our structure-aware generative model, when guided correctly, possesses a superior capacity to model complex chemical transformations even at scale, surpassing large-scale foundation models.

**Upper Bound with Oracle Reaction Centers.** A key advantage of our modular framework is that the generative backbone is decoupled from the reaction center predictor. To assess the full potential of our generative backbone, we evaluate performance using ground-truth (Oracle) reaction centers. We emphasize that the Oracle RC setting is included solely as an internal diagnostic to establish the capacity upper bound of our generative backbone and to identify the upstream RC predictor as the primary bottleneck (quantified

*Table 1.* **Top-$k$ Exact Match Accuracy on USPTO-50k** (Reaction Class Unknown). Baselines are categorized by paradigm. The best results are **bolded** and the second best are underlined.

| Model | Top-1 | Top-3 | Top-5 | Top-10 |
|---|---|---|---|---|
| ***LLM-based*** | | | | |
| RSGPT (Deng et al., 2025) | 63.4 | 84.2 | 89.2 | 93.0 |
| RSGPT ($\times$20) (Deng et al., 2025) | **77.0** | **90.9** | 94.3 | **96.7** |
| ***Template-based*** | | | | |
| RetroSim (Coley et al., 2017) | 37.3 | 54.7 | 63.6 | 74.1 |
| NeuralSym (Segler & Waller, 2017) | 44.4 | 65.3 | 72.4 | 78.9 |
| GLN (Dai et al., 2019) | 52.5 | 69.0 | 75.6 | 83.7 |
| LocalRetro (Chen & Jung, 2021) | 53.4 | 77.5 | 85.9 | 92.4 |
| RetroComposer (Yan et al., 2022) | 54.5 | 77.2 | 83.2 | 87.7 |
| ***Semi-template-based*** | | | | |
| G2G (Shi et al., 2020) | 48.9 | 67.6 | 72.5 | 75.5 |
| RetroXpert (Yan et al., 2020) | 50.4 | 61.1 | 62.3 | 63.4 |
| RetroPrime (Wang et al., 2021) | 51.4 | 70.8 | 74.0 | 76.1 |
| G$^2$Retro (Chen et al., 2023) | 54.1 | 74.1 | 81.2 | 86.7 |
| SemiRetro (Gao et al., 2022) | 54.9 | 75.3 | 80.4 | 84.1 |
| Graph2Edits (Zhong et al., 2023a) | 55.1 | 77.3 | 83.4 | 89.4 |
| ***Template-free*** | | | | |
| SCROP (Zheng et al., 2019) | 43.7 | 60.0 | 65.2 | 68.7 |
| MEGAN (Sacha et al., 2021) | 48.1 | 70.7 | 78.4 | 86.1 |
| Graph2SMILES (Tu & Coley, 2022) | 52.9 | 66.5 | 70.0 | 72.9 |
| Retroformer (Wan et al., 2022) | 53.2 | 71.1 | 76.6 | 82.1 |
| RetroBridge (Igashov et al., 2023) | 50.8 | 74.1 | 80.6 | 85.6 |
| Ualign (Zeng et al., 2024) | 53.6 | 77.6 | 84.6 | 90.3 |
| NAG2G (Yao et al., 2024) | 55.1 | 76.9 | 83.4 | 89.9 |
| R-SMILES (Zhong et al., 2022) | 56.3 | 79.2 | 86.2 | 91.0 |
| EditRetro (Han et al., 2024) | 60.8 | 80.6 | 86.0 | 90.3 |
| RetroProdFlow (Yadav et al., 2025) | 50.0 | 74.3 | 81.2 | 85.8 |
| RetroSynFlow (Yadav et al., 2025) | 60.0 | 77.9 | 82.7 | 85.3 |
| **Ours (Pred. RC)** | 61.2 | 81.5 | 86.2 | 89.2 |
| **Ours (Orac. RC)** | 71.1 | 90.8 | **94.5** | 96.0 |

*Table 2.* **Top-$k$ Exact Match Accuracy on USPTO-Full**. The best results are **bolded** and the second best are underlined.

| Model | Top-1 | Top-3 | Top-5 | Top-10 |
|---|---|---|---|---|
| ***LLM-based*** | | | | |
| RSGPT (Deng et al., 2025) | 59.2 | 74.2 | 78.2 | 82.1 |
| ***Template-based*** | | | | |
| RetroSim (Coley et al., 2017) | 32.8 | – | – | 56.1 |
| GLN (Dai et al., 2019) | 39.3 | – | – | 63.7 |
| LocalRetro (Chen & Jung, 2021) | 39.1 | 53.3 | 58.4 | 63.7 |
| ***Semi-template-based*** | | | | |
| RetroPrime (Wang et al., 2021) | 44.1 | 59.1 | 62.8 | 68.5 |
| ***Template-free*** | | | | |
| R-SMILES (Zhong et al., 2022) | 48.9 | 66.6 | 72.0 | 76.4 |
| NAG2G (Yao et al., 2024) | 49.7 | 64.6 | 69.3 | 74.0 |
| **Ours (Pred. RC)** | 51.3 | 67.8 | 72.3 | 75.8 |
| **Ours (Orac. RC)** | **63.4** | **77.6** | **80.9** | **83.6** |

systematically in Section 5.4); it is not intended as a practical deployment scenario, and the Pred. RC results should be used for direct comparison with other end-to-end methods. As shown in our analysis, under this Oracle setting, our method achieves a staggering 71.1% Top-1 accuracy on USPTO-50k. This performance significantly surpasses RSGPT (63.4%) and even approaches the performance of RSGPT with test-time augmentation (*RSGPT* $\times$20: 77.0%),

without requiring any test-time sampling tricks or massive pretraining. This result establishes a strong upper bound and confirms that our generative model is extremely capable; the primary bottleneck lies in the accuracy of the upstream reaction center predictor, which we analyze further in Section 5.4.

### 5.3. Analysis: Positional Inductive Bias vs. Scaling

A central hypothesis of this work is that encoding the two-stage structure of retrosynthesis as a positional inductive bias is more effective than scaling model parameters. To test this, we conducted experiments on USPTO-Full across four model sizes (280K, 2M, 8M, and 65M parameters).

**Effect of Ordering Strategies.** Figure 2 shows the Top-1, Top-3, and Top-5 accuracies (at 50 sampling steps) for models trained with *Canonical* and our *RC-Rooted* ordering strategies. The results show a clear trend: *RC-Rooted* ordering (green bars) consistently outperforms the *Canonical* baseline (blue bars) across all model scales. For instance, with the X-Large model (65M), our method achieves 51.2% Top-1 accuracy compared to 50.4% for Canonical.

More importantly, the *RC-Rooted (Oracle)* results (pink bars) provide strong support for our claim that structural bias outweighs scaling. Our Small model (280K parameters) with oracle ordering achieves $\approx$51% Top-1 accuracy, matching the X-Large model (65M parameters) with Canonical ordering (50.4%). This indicates that precise structural guidance can compensate for a 200$\times$ reduction in parameter count. Even with predicted reaction centers, we maintain a consistent advantage over Canonical ordering. The gap to oracle performance (reaching 63.4% at 65M) highlights the potential of the generative backbone when the predictor bottleneck is removed.

**Mechanism Verification: The Necessity of RoPE.** To understand *how* ordering improves performance, we analyze the impact of different positional embedding (PE) schemes on RetroDiT-8M (evaluated on USPTO-50k). As shown in Table 3, ordering atoms alone is insufficient if the network cannot exploit this sequence information. Without any PE (*w/o PE*), the model struggles to leverage the RC-Rooted structure, as the self-attention mechanism remains permutation invariant. While *Absolute PE* provides significant improvement, RoPE yields the highest accuracy across all metrics. This confirms that RoPE is essential for capturing the relative topological distances implied by our BFS-based ordering, effectively enabling the "Head-to-Tail" interaction between the reaction center and dummy nodes discussed in Section 4.2.

**Backbone Architecture.** Finally, our Diffusion Transformer (DiT) backbone consistently outperforms a standard Graph Transformer (GTF) across all model sizes. We at-

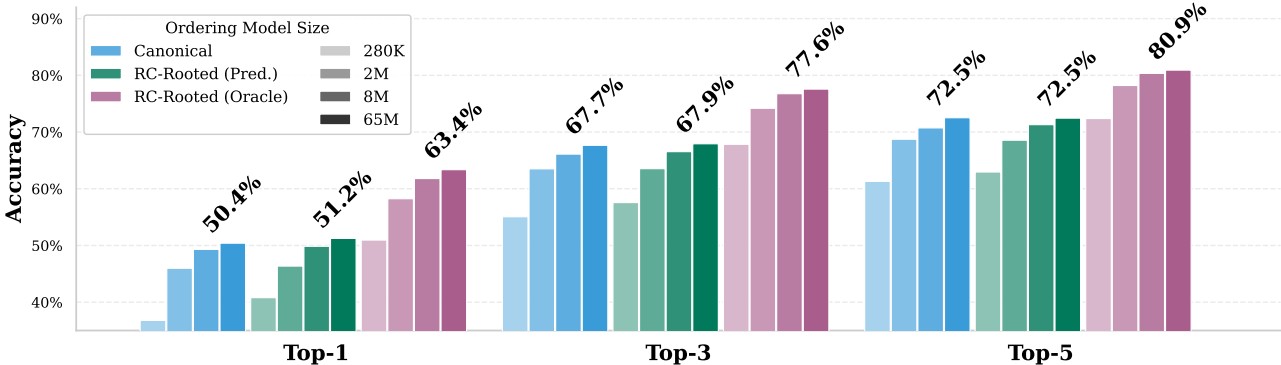

*Figure 2.* **Positional Inductive Bias vs. Model Scaling on USPTO-Full.** Performance comparison across four model sizes (280K to 65M). *RC-Rooted* ordering consistently outperforms the *Canonical* baseline. The Oracle setting with a Small model (280K) matches the Canonical X-Large model (65M), showing that structure-aware priors are more parameter-efficient than scaling alone.

*Table 3.* **Ablation of Positional Embeddings.** Performance of RetroDiT-8M with RC-Rooted ordering under different PE configurations on USPTO-50k.

| Configuration | Top-1 (%) | Top-5 (%) | Top-10 (%) |
|---|---|---|---|
| w/o PE | 50.4 | 77.3 | 81.5 |
| w/ Absolute PE | 68.5 | 93.2 | 94.8 |
| **w/ RoPE (Ours)** | **71.0** | **94.6** | **96.5** |

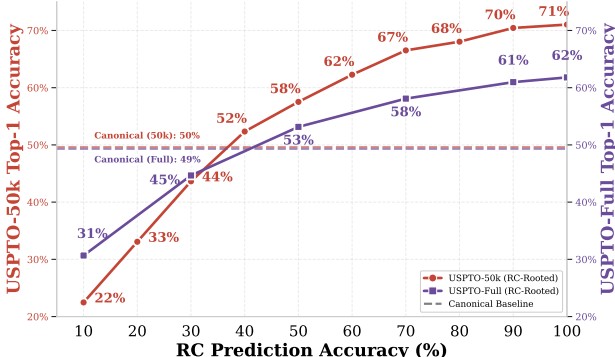

*Figure 3.* **Impact of RC Prediction Accuracy on Generation Performance.** The solid lines track the Top-1 Accuracy of our RC-Rooted model as the reaction center accuracy varies from 10% to 100%. The dashed lines represent the performance of the Canonical ordering baseline (which is independent of RC prediction).

tribute this to DiT's adaptive layer normalization, which better handles the time-dependent dynamics of the flow matching process. Detailed comparisons are provided in Appendix F.2.

**5.4. The Bottleneck: Impact of RC Prediction Accuracy**

Our modular framework allows us to isolate the impact of reaction center (RC) prediction on overall performance. To quantify this impact, we conducted a sensitivity analysis on both USPTO-50k and USPTO-Full. Using fixed RetroDiT-8M models trained with RC-rooted ordering, we simulated inference with varying RC accuracies from 10% to 100% by randomly replacing the correct root with a random atom with probability $(1 - \text{Acc}\%)$.

**Sensitivity Analysis.** Figure 3 shows the Top-1 generation accuracy as a function of simulated RC accuracy. The results reveal two key observations. First, generation accuracy exhibits a strong positive correlation with RC accuracy on both datasets. As RC accuracy approaches 100%, generation accuracy reaches 71% on USPTO-50k and 62% on USPTO-Full, establishing clear upper bounds. Second, we observe a crossover point with the Canonical baseline (dashed lines) at approximately 35–40% RC accuracy. Below this threshold, incorrect roots actively mislead the model, resulting in worse performance than permutation-invariant canonical ordering. Above this threshold, the structural inductive bias yields consistent gains, reaching over 10 points improvement at 70% RC accuracy.

**Implications.** This analysis confirms a key finding: the generative backbone is not the primary performance bottleneck. RetroDiT is effective at reconstructing reactants when conditioned on correct structural context. The performance gap between our predicted setting ($\approx$61% on USPTO-50k) and the oracle upper bound (71%) is largely due to the limited accuracy of the current RC predictor. This validates our modular design: the framework provides a strong generative backbone that can directly benefit from future advances in RC prediction without requiring retraining of the generator. Additional analysis is provided in Appendix F.4.

## 6. Conclusion and Future Work

We propose a structure-aware template-free paradigm for single-step retrosynthesis that encodes the two-stage nature of chemical reactions as a positional inductive bias through reaction-center-rooted atom ordering. Combined with RetroDiT and discrete flow matching, our approach combines the structural precision of semi-template methods with the generative flexibility of template-free approaches.

On USPTO-50k and USPTO-Full, our method achieves state-of-the-art performance while training 6× faster and requiring only 20–50 sampling steps compared to 500 for prior diffusion methods. Experiments with oracle reaction centers establish upper bounds of 71.1% and 63.4% Top-1 accuracy, showing that the generative backbone is effective.

Beyond these results, our work highlights a broader principle: ***well-designed inductive biases outperform brute-force scaling***. A 280K-parameter model with proper ordering matches a 65M-parameter model without it, demonstrating that domain-specific structural priors can be more effective than increasing model size or training data and suggesting a promising direction for efficient and accurate molecular design in AI for Science. Future work will focus on improving reaction center prediction and extending this framework to multi-step retrosynthetic planning.

## Acknowledgments

This work was supported by the Shanghai Municipal Special Program for Basic Research on General AI Foundation Models (Grant No. 2025SHZDZX025D05), in collaboration with Shanghai Artificial Intelligence Laboratory. We acknowledge Shanghai Artificial Intelligence Laboratory for providing computational resources.

## Impact Statement

This paper presents work whose goal is to advance the field of Machine Learning. There are many potential societal consequences of our work, none which we feel must be specifically highlighted here.

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

## A. Detailed Definition and Extraction of Reaction Centers

In our framework, the accurate identification of reaction centers is crucial for constructing the positional inductive bias. We implement a rigorous extraction algorithm using RDKit to identify all atoms in the product molecule that undergo chemical changes during the reaction. Specifically, we classify reaction centers into two major groups covering eight distinct types of chemical transformations.

Let $G_P$ and $G_R$ denote the product and reactant graphs, respectively. An atom $a$ (identified by its atom-mapping number) is considered part of the reaction center if it satisfies any of the following criteria:

**Topological Changes**    These criteria capture changes in the connectivity or bond types associated with an atom:

1. **Formed Bonds**: Atom $a$ is part of a bond $(a, b)$ that exists in $G_P$ but does not exist in $G_R$. This corresponds to bond formation in the forward reaction (or bond breaking in retrosynthesis).

2. **Broken Bonds**: Atom $a$ is part of a bond $(a, b)$ that exists in $G_R$ but does not exist in $G_P$. This corresponds to bond breaking in the forward reaction (or bond formation in retrosynthesis).

3. **Bond Order Changes**: Atom $a$ is part of a bond $(a, b)$ that exists in both $G_P$ and $G_R$, but its bond type (e.g., single, double, triple, aromatic) differs.

**Atom Property Changes**    These criteria capture changes in the intrinsic chemical properties of an atom, even if its local connectivity remains unchanged:

4. **Charge Changes**: The formal charge of atom $a$ differs between $G_P$ and $G_R$.

5. **H-Count Changes**: The total number of implicit and explicit hydrogen atoms attached to atom $a$ changes (e.g., due to protonation/deprotonation).

6. **Chirality Changes**: The chiral tag of atom $a$ changes (e.g., inversion of stereochemistry from $R$ to $S$, or loss/gain of a chiral center).

7. **Aromaticity Changes**: The aromatic state of atom $a$ changes (e.g., due to ring opening or closure).

8. **Hybridization Changes**: The orbital hybridization of atom $a$ changes (e.g., $sp^3 \rightarrow sp^2$).

### A.1. Extraction Logic via Atom Mapping

Our extraction script systematically compares the product and reactant SMILES based on atom mapping numbers. If any of the above eight changes are detected for an atom, it is flagged as a reaction center. This fine-grained definition ensures that our model captures subtle chemical transformations that might be missed by coarser definitions. The specific implementation using RDKit is provided in Listing 1. This function returns a dictionary containing lists of product atom indices for each of the eight change categories.

```python
from rdkit import Chem

def get_reaction_center(rxn_smiles):
    reactants, _, products = rxn_smiles.split('>')
    reactant_mols = [Chem.MolFromSmiles(r) for r in reactants.split('.') if r]
    product_mols = [Chem.MolFromSmiles(p) for p in products.split('.') if p]

    # 1. Build Map: Atom Map Number -> Product Atom Index
    product_map_to_idx = {}
    for mol in product_mols:
        if mol is None: continue
        for atom in mol.GetAtoms():
            map_num = atom.GetAtomMapNum()
            if map_num: product_map_to_idx[map_num] = atom.GetIdx()

    product_atom_maps = set(product_map_to_idx.keys())
```

```
17    rc_dict = {k: [] for k in ['formed_bonds', 'broken_bonds', 'bond_order_changes',
18                                'charge_changes', 'h_count_changes', 'chirality_changes',
19                                'aromatic_changes', 'hybridization_changes']}
20
21    # Helper to extract bond dict: (map1, map2) -> bond_type
22    def get_bond_info(mols):
23        bonds = {}
24        for mol in mols:
25            if mol is None: continue
26            for bond in mol.GetBonds():
27                m1 = bond.GetBeginAtom().GetAtomMapNum()
28                m2 = bond.GetEndAtom().GetAtomMapNum()
29                if m1 and m2:
30                    bonds[tuple(sorted((m1, m2)))] = bond.GetBondTypeAsDouble()
31        return bonds
32
33    r_bonds = get_bond_info(reactant_mols)
34    p_bonds = get_bond_info(product_mols)
35
36    # 2. Detect Topological Changes
37    # Formed Bonds (in Product but not Reactant)
38    for bond in set(p_bonds.keys()) - set(r_bonds.keys()):
39        if bond[0] in product_atom_maps and bond[1] in product_atom_maps:
40            rc_dict['formed_bonds'].append([product_map_to_idx[bond[0]],
41                                            product_map_to_idx[bond[1]]])
42
43    # Broken Bonds (in Reactant but not Product)
44    for bond in set(r_bonds.keys()) - set(p_bonds.keys()):
45        # Only record atoms that still exist in the product
46        indices = []
47        if bond[0] in product_map_to_idx: indices.append(product_map_to_idx[bond[0]])
48        if bond[1] in product_map_to_idx: indices.append(product_map_to_idx[bond[1]])
49        if indices: rc_dict['broken_bonds'].append(indices)
50
51    # Bond Order Changes
52    for bond in set(r_bonds.keys()) & set(p_bonds.keys()):
53        if r_bonds[bond] != p_bonds[bond]:
54            if bond[0] in product_atom_maps and bond[1] in product_atom_maps:
55                rc_dict['bond_order_changes'].append([product_map_to_idx[bond[0]],
56                                                      product_map_to_idx[bond[1]]])
57
58    return rc_dict
```

*Listing 1.* Algorithm for extracting 8 types of reaction centers from atom-mapped SMILES.

### A.2. Dataset-level Analysis

Having defined the eight specific categories of reaction centers, we now analyze their distribution across the datasets to understand the prevalence of different chemical phenomena and data characteristics.

**Distribution of Reaction Center Types.** Figure 4 shows the percentage of reactions containing each type of chemical change across training, validation, and test splits for both USPTO-50k and USPTO-Full. The distributions reveal significant structural differences between the two datasets:

- **USPTO-50k** (Figure 4a): H-count changes are nearly universal ($> 91\%$), reflecting the ubiquity of proton transfer. Bond breaking is the second most prevalent ($> 93\%$), while formed bonds occur in 71.5% of reactions. Notably, bond order changes are relatively sparse ($\approx 11\%$), and subtle property changes like chirality ($\approx 6\%$) and aromaticity ($\approx 2\%$) remain rare.

- **USPTO-Full** (Figure 4b): A striking distributional shift emerges. While H-count changes remain dominant ($> 90\%$), the pattern reverses for topological changes: formed bonds are now most prevalent (78.6%), bond breaking becomes

extremely sparse (≈2.2%), and bond order changes play a substantially larger role (≈32%). This suggests USPTO-Full contains different reaction types compared to USPTO-50k.

- **Consistency across splits**: All eight change types show remarkably stable distributions across train, validation, and test splits in both datasets, with variations typically under 1%, confirming proper dataset partitioning.

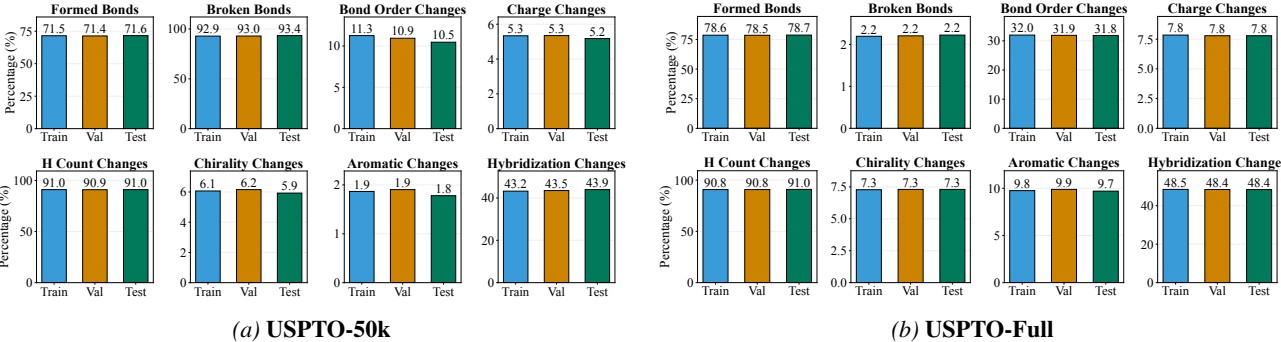

*(a)* **USPTO-50k**                     *(b)* **USPTO-Full**

*Figure 4.* **Distribution of Reaction Center Types.** Percentage of reactions containing each chemical change type. Note the distinct distributional shifts between datasets, particularly in bond formation/breaking patterns.

**Coverage Analysis.** To assess the completeness of our eight-category definition, Figure 5 plots the cumulative percentage of reactions covered by the top-$k$ change types (ordered by frequency for each dataset). Despite the combinatorial complexity of chemical reactions, coverage saturates remarkably quickly. The top-1 change type alone covers over 90% of reactions in both datasets. By including just the top-2 types, cumulative coverage exceeds **99%** across all splits. Specifically:

- **USPTO-50k**: The top-2 types (H-count changes and bond breaking) achieve >99% coverage, with the curve plateauing by the third type.

- **USPTO-Full**: Similarly, the top-2 types (H-count changes and formed bonds) cover >99% of reactions, demonstrating that despite the distributional shift, the eight-category framework remains comprehensive.

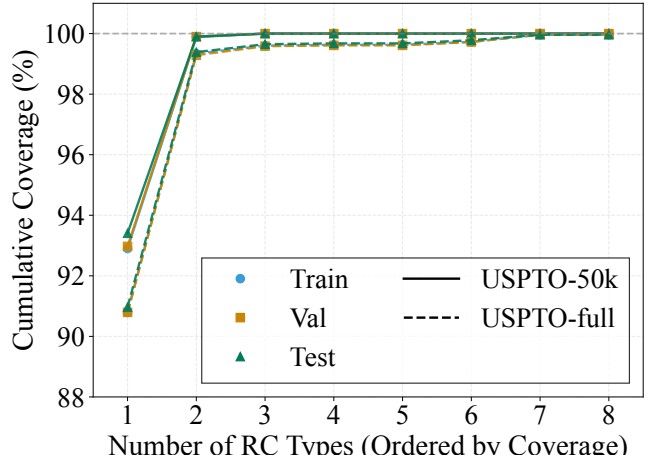

*Figure 5.* **Cumulative Reaction Coverage.** Change types ordered by prevalence. The top-2 types alone cover over 99% of reactions in both datasets.

This analysis confirms that our reaction center definition is both chemically intuitive and statistically complete, capturing effectively all reaction patterns in standard benchmarks while revealing important dataset characteristics that may influence model generalization. Critically, this comprehensive coverage addresses a key limitation of prior work. Semi-template methods (Shi et al., 2020) and synthon-based approaches (Yadav et al., 2025) typically consider only two transformation types: formed bonds and broken bonds. While sufficient for USPTO-50k (where these two types have high coverage), this simplified definition fails on USPTO-Full, covering at most 80.8% (78.6% formed + 2.2% broken) of reactions. The remaining 20% involve bond order changes, hybridization changes, and other property modifications that are ignored by these approaches. Our eight-category framework overcomes this limitation, achieving >99% coverage with just the top-2 types across both datasets, ensuring robust performance regardless of reaction type or dataset distribution.

# B. Implementation of Rooted Atom Ordering

To enforce the positional inductive bias, we leverage the robust canonicalization routines provided by RDKit. The core mechanism relies on the `rootedAtAtom` argument in the SMILES generation function, which forces the traversal to

begin at a specific atom index (assigning it Index 0 in the sequence) while determining the order of the remaining atoms deterministically based on canonical invariants (e.g., Morgan algorithm).

We implement two specific variations of this routine for the training and inference phases, respectively.

**Training Phase.** The training data typically contains atom-mapped SMILES. To ensure the model learns from the intrinsic chemical structure rather than arbitrary mapping numbers, we first strip the atom map information before generating the rooted sequence.

```python
from rdkit import Chem

def Canonicalize(smi, root=-1):
    """
    Clears atom map numbers and generates a canonical/rooted SMILES.
    Args:
        smi (str): Input SMILES with atom maps.
        root (int): Index of the atom to set as the root (Index 0).
                    If -1, standard canonicalization is used.
    """
    mol = Chem.MolFromSmiles(smi)
    # Clear existing atom maps to ensure clean features
    for atom in mol.GetAtoms():
        if atom.HasProp('molAtomMapNumber'):
            atom.ClearProp('molAtomMapNumber')

    # Generate SMILES rooted at the specific atom
    return Chem.MolToSmiles(mol, isomericSmiles=True, rootedAtAtom=root, canonical=True)
```

*Listing 2.* Rooted SMILES generation during training (with map clearing).

**Inference Phase.** During inference, the input is a standard product SMILES without ground-truth atom maps. The reaction center predictor outputs the index of the predicted root atom. We directly use this index to reorder the graph.

```python
def Canonicalize(smi, root_idx):
    """
    Generates a SMILES string rooted at the predicted reaction center.
    Args:
        smi (str): Input Product SMILES (canonical).
        root_idx (int): The index of the predicted reaction center atom.
    """
    mol = Chem.MolFromSmiles(smi)
    # Force the traversal to start at root_idx
    # The rest of the graph is ordered canonically by RDKit
    return Chem.MolToSmiles(mol, rootedAtAtom=root_idx, canonical=True)
```

*Listing 3.* Rooted SMILES generation during inference.

This implementation ensures that for any given reaction center, the resulting sequence of atoms is unique, deterministic, and chemically meaningful, providing a stable input for the RetroDiT backbone.

## C. Probabilistic Decomposition and Training Objectives

In this appendix, we provide a detailed derivation of the conditional independence assumptions underlying our discrete flow matching framework for retrosynthesis, and formally present the node-level and edge-level training objectives.

### C.1. Conditional Independence Decomposition

Our framework operates on molecular graphs $G = (X, E)$, where $X = \{x^{(n)}\}_{n=1}^{N}$ denotes the set of node (atom) features with $x^{(n)} \in \mathcal{X}$, and $E = \{e^{(ij)}\}_{1 \leq i < j \leq N}$ denotes the set of edge (bond) features with $e^{(ij)} \in \mathcal{E}$. Following standard conventions (Vignac et al., 2022), we include a "no-edge" category in $\mathcal{E}$ to represent the absence of bonds.

**Factorization of the Noising Process.** We adopt the linear interpolation noising process from DeFoG (Qin et al., 2024). The conditional probability of observing an intermediate graph $G_t$ given the clean target graph $G_1$ factorizes over individual nodes and edges under a conditional independence assumption:

$$p_{t|1}(G_t|G_0, G_1) = \prod_{n=1}^{N} p_{t|1}\left(x_t^{(n)}|x_0^{(n)}, x_1^{(n)}\right) \prod_{1 \le i < j \le N} p_{t|1}\left(e_t^{(ij)}|e_0^{(ij)}, e_1^{(ij)}\right), \tag{7}$$

where each component follows the linear interpolation path:

$$p_{t|1}\left(x_t^{(n)}|x_0^{(n)}, x_1^{(n)}\right) = t \cdot \delta\left(x_t^{(n)}, x_1^{(n)}\right) + (1-t) \cdot \delta\left(x_t^{(n)}, x_0^{(n)}\right), \tag{8}$$

$$p_{t|1}\left(e_t^{(ij)}|e_0^{(ij)}, e_1^{(ij)}\right) = t \cdot \delta\left(e_t^{(ij)}, e_1^{(ij)}\right) + (1-t) \cdot \delta\left(e_t^{(ij)}, e_0^{(ij)}\right), \tag{9}$$

where $\delta(\cdot, \cdot)$ denotes the Kronecker delta function.

**Factorization of the Denoising Prediction.** To enable tractable inference, we assume that the learned denoising network $f_\theta$ predicts the marginal distributions of the clean graph $G_1$ given an intermediate graph $G_t$ in a factorized form:

$$p_{1|t}^\theta(\cdot|G_t) = \left(\left\{p_{1|t}^{\theta,(n)}(\cdot|G_t)\right\}_{n=1}^{N}, \left\{p_{1|t}^{\theta,(ij)}(\cdot|G_t)\right\}_{1 \le i < j \le N}\right), \tag{10}$$

where $p_{1|t}^{\theta,(n)}(\cdot|G_t) \in \Delta^{|\mathcal{X}|}$ and $p_{1|t}^{\theta,(ij)}(\cdot|G_t) \in \Delta^{|\mathcal{E}|}$ represent the predicted probability distributions over node and edge categories, respectively. Here, $\Delta^K$ denotes the $(K-1)$-dimensional probability simplex.

**Remark on Conditional Independence.** We emphasize that while the noising process and denoising prediction adopt factorized forms, the intermediate graph $G_t$ serves as the shared conditioning variable. This allows the neural network $f_\theta$ to capture complex dependencies across nodes and edges through its architecture (e.g., graph transformers), while maintaining computational tractability. The factorization is applied only to the output probability distributions, not to the internal representations.

## C.2. Node-Level and Edge-Level Training Objectives

Given the factorization in Eq. (10), the training objective decomposes into independent cross-entropy losses over nodes and edges.

**Per-Component Cross-Entropy Loss.** For a single training pair $(G_0, G_1)$ representing the product-to-reactant transformation, sampled time $t \sim \mathcal{U}(0, 1)$, and intermediate graph $G_t \sim p_{t|0,1}(G_t|G_0, G_1)$, the cross-entropy loss is defined as:

$$\text{CE}(G_1, p_{1|t}^\theta(\cdot|G_t)) = \mathcal{L}_{\text{node}} + \lambda \cdot \mathcal{L}_{\text{edge}}, \tag{11}$$

where $\lambda \in \mathbb{R}^+$ is a weighting hyperparameter that balances the contribution of node and edge losses. The node-level and edge-level losses are given by:

$$\mathcal{L}_{\text{node}} = -\sum_{n=1}^{N} \log p_{1|t}^{\theta,(n)}\left(x_1^{(n)}|G_t\right), \tag{12}$$

$$\mathcal{L}_{\text{edge}} = -\sum_{1 \le i < j \le N} \log p_{1|t}^{\theta,(ij)}\left(e_1^{(ij)}|G_t\right). \tag{13}$$

**Full Training Objective.** The complete training objective minimizes the expected cross-entropy over the data distribution, time distribution, and noising process:

$$\mathcal{L}_{\text{Retro}}(\theta) = \mathbb{E}_{t,(G_0,G_1),G_t}\left[-\sum_{n=1}^{N} \log p_{1|t}^{\theta,(n)}\left(x_1^{(n)}|G_t\right) - \lambda \sum_{1 \le i < j \le N} \log p_{1|t}^{\theta,(ij)}\left(e_1^{(ij)}|G_t\right)\right]. \tag{14}$$

**Connection to Rate Matrix Estimation.** Following the theoretical analysis in DeFoG (Qin et al., 2024), minimizing the cross-entropy loss in Eq. (14) is directly related to minimizing the estimation error of the rate matrices used during inference. Specifically, for each node $n$ and edge $(i, j)$, the corresponding rate matrices are computed as:

$$R_t^{(n)}\left(x_t^{(n)}, x'|x_0^{(n)}\right) = \mathbb{E}_{x_1^{(n)} \sim p_{1|t}^{\theta,(n)}(\cdot|G_t)}\left[R_t\left(x_t^{(n)}, x'|x_0^{(n)}, x_1^{(n)}\right)\right], \tag{15}$$

$$R_t^{(ij)}\left(e_t^{(ij)}, e'|e_0^{(ij)}\right) = \mathbb{E}_{e_1^{(ij)} \sim p_{1|t}^{\theta,(ij)}(\cdot|G_t)}\left[R_t\left(e_t^{(ij)}, e'|e_0^{(ij)}, e_1^{(ij)}\right)\right], \tag{16}$$

where $R_t(\cdot, \cdot|\cdot, \cdot)$ is the conditional rate matrix defined in Eq. (6) of the main text. The factorized prediction allows for efficient computation of these rate matrices during the CTMC-based sampling procedure.

## D. RetroDiT Architecture

The detailed internal structure of our RetroDiT transformer block is illustrated in Figure 6. The architecture is designed to process node and edge embeddings in parallel while fostering deep interaction between them:

1. **Node Path with RoPE**: The node features undergo Masked Multi-Head Attention. Crucially, RoPE are injected at this stage to modulate the query and key vectors, enforcing the reaction-center-rooted positional inductive bias.

2. **Edge-Conditioned Attention**: Edge embeddings are processed through a Feed-Forward Network (FFN) to generate "Shift and Scale" parameters. These parameters directly modulate the attention scores matrix, allowing bond types (or the absence thereof) to gate the information flow between atoms.

3. **Dual Updates**: Both node and edge representations are subsequently updated through their respective Feed-Forward Networks, ensuring that structural changes propagate through both atomic and bonding features.

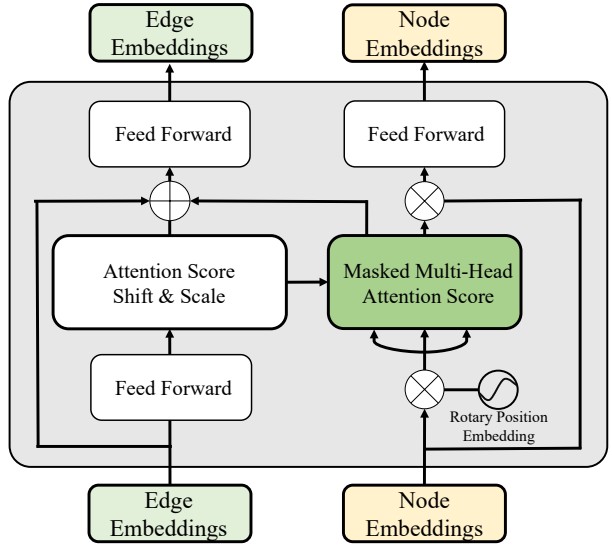

*Figure 6.* **RetroDiT: Graph Transformer with Rotary Position Embedding.**

## E. Experimentation Details

### E.1. Details on Large-Scale Baselines (RSGPT)

In our main results, RSGPT (Deng et al., 2025) serves as a powerful upper-bound baseline representing the "scaling law" approach. To contextualize the efficiency comparison, we highlight the substantial disparity in resource requirements and complexity between RSGPT and our framework:

- **Data Scale (10 Billion vs. 50k)**: RSGPT relies on a massive synthetic dataset generation process. It utilizes RDChiral to extract templates and applies them to molecular fragments from PubChem, ChEMBL, and Enamine, generating over 10 billion synthetic reaction data points for pre-training. In contrast, our standard model is trained only on the original 50,000 labeled reactions from USPTO-50k.

- **Model Scale (3.2B vs. ∼65M)**: RSGPT is built upon the LLaMA2 architecture with approximately 3.2 billion parameters (24 layers, 2048 hidden dimensions). Our largest RetroDiT model (X-Large) contains only 65 million parameters, making it approximately $50\times$ smaller while achieving competitive performance.

- **Training Complexity (Multi-stage vs. End-to-End)**: The training pipeline of RSGPT is highly complex, involving three distinct stages: (1) Large-scale Pre-training on synthetic data; (2) Reinforcement Learning from AI Feedback

(RLAIF) (Lee et al., 2023; Li et al., 2024; Lee et al., 2024) using Proximal Policy Optimization (Schulman et al., 2017) to align with chemical validity; and (3) Supervised Fine-tuning. Our method employs a streamlined, simulation-free Discrete Flow Matching objective that converges in a single training stage.

- **Inference Protocol (Beam Search vs. Sampling)**: Regarding inference, RSGPT employs a deterministic Beam Search strategy, often combined with $20\times$ Test-Time Augmentation to boost performance. Notably, the specific beam size parameter, a critical factor for inference cost and performance, is not explicitly disclosed in their documentation or supplementary materials. In contrast, our method adheres to the standardized stochastic sampling protocol ($N = 100$ independent samples) used by previous generative baselines (Igashov et al., 2023; Maziarz et al., 2025), ensuring transparency and reproducibility.

This comparison underscores that while RSGPT achieves SOTA results through massive resource utilization and complex, partially opaque inference heuristics, our approach achieves comparable results through structural inductive biases, algorithmic efficiency, and a transparent evaluation protocol.

### E.2. Reaction Center Predictor

Our reaction center predictor is designed to identify atoms and bonds that undergo chemical changes during retrosynthesis, as defined by the eight-category framework in Appendix A. Unlike prior semi-template approaches that focus exclusively on bond formation and breaking (Shi et al., 2020), our predictor handles the full spectrum of topological and property changes, ensuring comprehensive coverage across diverse reaction types.

**Problem Formulation.** We formulate reaction center prediction as a unified binary classification task over both edges (bonds) and nodes (atoms) in the product molecular graph. Given a product graph $G_p = (A, X)$ with adjacency matrix $A$ and node features $X$, the predictor outputs reactivity scores for all bonds and atoms, indicating which structural elements will undergo changes.

**Model Architecture.** Following (Shi et al., 2020), we employ a Relational Graph Convolutional Network (R-GCN) (Schlichtkrull et al., 2018) as the graph encoder to compute node embeddings $H^L \in \mathbb{R}^{n \times k}$ and graph-level embedding $h_{G_p} \in \mathbb{R}^k$ from the product graph. Building upon this foundation, we extend the architecture with a dual-branch scoring mechanism to handle both bond-level and atom-level reaction centers:

- **Edge-level scoring**: For each bond $(i, j)$ in $G_p$, we construct an edge embedding by concatenating the node embeddings of atoms $i$ and $j$, the bond type feature $A_{ij}$, and the global graph embedding $h_{G_p}$:

$$e_{ij} = H_i^L \| H_j^L \| A_{ij} \| h_{G_p} \tag{17}$$

  The reactivity score is computed as $s_{ij}^{\text{edge}} = \sigma(m_e(e_{ij}))$, where $m_e(\cdot)$ is a feedforward network and $\sigma(\cdot)$ denotes the sigmoid function.

- **Node-level scoring**: For each atom $i$ in $G_p$, we construct a node embedding by concatenating its node embedding with the global graph context:

$$v_i = H_i^L \| h_{G_p} \tag{18}$$

  The reactivity score is computed as $s_i^{\text{node}} = \sigma(m_n(v_i))$, where $m_n(\cdot)$ is a feedforward network.

The inclusion of global graph embeddings in both branches enables the model to capture long-range dependencies and reaction context beyond local neighborhoods, which is essential for identifying reactive sites influenced by distant functional groups.

**Training Strategy.** During training, we adopt a hierarchical labeling strategy to generate ground-truth supervision: we prioritize bond-level changes over atom-level changes. Specifically, for each reaction in the training data, we first check whether any bonds undergo topological changes according to the eight-category definition in Appendix A. If bond-level reaction centers exist, we train only the edge-level branch with these bond labels; otherwise, we train only the node-level branch with atom-level property change labels. This ensures that the model learns to prioritize bond-level reactivity while maintaining the capability to handle reactions involving purely atom property modifications.

The predictor is trained using weighted binary cross-entropy loss to address the severe class imbalance inherent in reaction center prediction:

$$\mathcal{L}_{\text{RC}} = \begin{cases} -\sum_{(i,j)} \lambda_e Y_{ij}^{\text{edge}} \log(s_{ij}^{\text{edge}}) + (1 - Y_{ij}^{\text{edge}}) \log(1 - s_{ij}^{\text{edge}}) & \text{if bond-level RCs exist} \\ -\sum_i \lambda_n Y_i^{\text{node}} \log(s_i^{\text{node}}) + (1 - Y_i^{\text{node}}) \log(1 - s_i^{\text{node}}) & \text{otherwise} \end{cases} \tag{19}$$

where $Y_{ij}^{\text{edge}}$ and $Y_i^{\text{node}}$ are binary ground-truth labels, and $\lambda_e, \lambda_n > 1$ are weighting hyperparameters that upweight positive examples. Ground-truth labels are automatically extracted from atom-mapped reaction data in USPTO-50k and USPTO-Full by comparing product and reactant molecular graphs according to the criteria in Appendix A.

**Inference Procedure.** At inference time, we unify bond-level and atom-level predictions into a single node-level scoring scheme for consistent comparison and selection:

1. **Score computation**: Compute reactivity scores for all bonds $s_{ij}^{\text{edge}}$ and all atoms $s_i^{\text{node}}$ in $G_p$.

2. **Score propagation**: For each atom $i$, aggregate bond-level scores to node level by summing scores from all bonds connected to atom $i$:

$$s_i^{\text{unified}} = s_i^{\text{node}} + \sum_{j \in \mathcal{N}(i)} s_{ij}^{\text{edge}} \tag{20}$$

   where $\mathcal{N}(i)$ denotes the set of neighboring atoms of atom $i$.

3. **Selection**: Select the top-$k$ atoms with the highest unified scores $s_i^{\text{unified}}$ as predicted reaction centers.

This unified scoring approach naturally prioritizes bond-level changes (since edge scores propagate to nodes) while ensuring that atom-level property changes are also considered. The predicted reaction centers are then used as root nodes for RDKit-based atom ordering, which provides the complete node sequence for RetroDiT's positional inductive bias. By handling all eight categories of chemical changes through this unified framework, our predictor avoids the coverage limitations of bond-only prediction methods (Shi et al., 2020), particularly for datasets like USPTO-Full where bond order changes and other property modifications are prevalent.

### E.3. Evaluation Metrics and Protocol

We provide a detailed description of the metrics used to evaluate our retrosynthesis framework.

**Top-$k$ Exact Match Accuracy.** This metric measures whether the ground-truth reactants are present within the model's top-$k$ predictions. Following the standard protocol (Maziarz et al., 2025), for each test product, we sample $M = 100$ candidate reactant sets using our generative model. These candidates are then canonicalized and deduplicated. We rank them based on the model's likelihood (frequency in the sampled set) and check if the canonical SMILES of the ground-truth reactants appear in the top $k$ ranks ($k \in \{1, 3, 5, 10\}$).

**Round-Trip Metrics.** To assess the chemical feasibility of the generated reactants, we employ a "round-trip" verification strategy. We use a pretrained ReactionT5 (Sagawa & Kojima, 2025) as a forward synthesis oracle to predict the product from our generated reactants.

- **Round-Trip Accuracy**: The percentage of test cases where the top-1 predicted reactant set, when fed into the forward model, successfully reproduces the original target product. This metric serves as a proxy for the chemical validity of the model's most confident prediction.

- **Round-Trip Coverage**: The percentage of test cases where at least one of the generated candidate reactant sets successfully reproduces the target product. This metric evaluates the diversity and the ability of the model to find *any* valid synthetic pathway, even if it is not the ground truth.

For the space limit in the main text, we leave the detailed results of this part in Appendix F.1 (Chemical Validity and Consistency).

### E.4. Hyperparameters and Implementation Details

**Data Preprocessing.** Following the protocols established by RetroBridge (Igashov et al., 2023) and RetroFlow (Yadav et al., 2025), we filter the USPTO-50k and USPTO-Full datasets to include only reactions where both reactants and products contain no more than 100 atoms. To accommodate leaving groups during the generative process, we append a fixed number of dummy nodes to the product graphs: $K = 10$ for USPTO-50k and $K = 20$ for USPTO-Full.

The discrete state spaces for nodes and edges are defined as follows:

- **Edge Types**: Both datasets utilize 4 bond types: Single, Double, Triple, and Aromatic. An additional "No Bond" type is implicit.

- **Node Types of USPTO-50k**: 17 types including {C, O, N, F, Cl, S, Br, B, I, Si, P, Sn, Mg, Zn, Cu, Se} and a special token "*" (empty).

- **Node Types of USPTO-Full**: 45 types including {C, O, N, F, Cl, S, Br, Si, I, B, P, Mg, Sn, Li, Zn, Cu, Se, Mn, Al, Na, Cr, K, H, Sb, Ge, Te, As, Pb, Ti, Hg, Zr, Bi, Ru, Ta, Xe, Pt, Ga, Ag, Fe, Mo, Co, W, Tl, Ca} and "*".

**Feature Representation and Alignment.** We represent the discrete node and edge types as one-hot vectors based on the vocabularies defined above. Crucially, to facilitate the conditional flow matching process, we enforce a strict index alignment between the product and reactant graphs. Since the atoms in the product molecule form a subset of the atoms in the reactants, we align the node features such that the first $N_{prod}$ nodes in the reactant graph correspond one-to-one with the atoms in the ordered product graph. The subsequent nodes in the reactant graph correspond to the leaving groups, which map to the initialized dummy nodes in the source (product) graph.

**Model Architectures.** To analyze the impact of model scaling, we instantiated our RetroDiT backbone in four different sizes ranging from 280k to 65.6M parameters. The specific configurations for the hidden node dimension ($d_{\text{node}}$), edge dimension ($d_{\text{edge}}$), number of layers ($L$), and attention heads ($H$) are detailed in Table 4.

*Table 4.* **RetroDiT Model Configurations.** Architecture details for the four model variants used in the scaling analysis.

| Model Variant | Params | Node Dim ($d_{\text{node}}$) | Edge Dim ($d_{\text{edge}}$) | Layers ($L$) | Heads ($H$) |
|---|---|---|---|---|---|
| Small | 280k | 64 | 32 | 4 | 2 |
| Medium | 2M | 128 | 64 | 6 | 4 |
| Large | 8M | 256 | 128 | 8 | 4 |
| X-Large | 65M | 512 | 256 | 16 | 8 |

**Training Configuration.** We train the models with a global batch size of 64, using mixed-precision training (`bf16-mixed`). The time step $t$ is sampled from a uniform distribution $t \sim \mathcal{U}[0, 1]$. The total loss is a weighted sum of node and edge cross-entropy losses, defined as $\mathcal{L} = \mathcal{L}_{\text{node}} + \lambda \mathcal{L}_{\text{edge}}$, where $\lambda = 10$ for USPTO-50k and $\lambda = 20$ for USPTO-Full.

We use the AdamW optimizer (Loshchilov & Hutter, 2019) with `amsgrad=True`. The hyperparameters are set as follows: peak learning rate $5 \times 10^{-4}$, weight decay $1 \times 10^{-12}$, $\beta_1 = 0.9$, $\beta_2 = 0.999$, and gradient clipping with max norm 1.0. We employ a Polynomial LR Scheduler:

- **Warmup**: Linear warmup for the first 500 steps starting from $1 \times 10^{-7}$.

- **Decay**: Polynomial decay with power $p = 0.3$ down to a final learning rate of $1 \times 10^{-5}$.

Validation is performed every 256 steps for USPTO-50k and every 512 steps for USPTO-Full. All models were trained on 8 NVIDIA GPUs, except for the 65M-parameter model on USPTO-Full, which was trained on 16 GPUs with a reduced batch size of 32.

**Inference Configuration.** During sampling, we employ an Euler discretization scheme with uniform time steps over the interval $[0, 1]$. For our main results, we use $N = 50$ steps (step size $\Delta t = 0.02$).

# F. Additional Experimental Results

This appendix provides comprehensive experimental results that were omitted from the main text due to space constraints. We present detailed performance breakdowns across different model sizes, ordering strategies, sampling steps, and backbone architectures.

## F.1. Detailed Performance on USPTO-50k

In this section, we provide the complete performance metrics for our systematic study on the USPTO-50k dataset. We evaluate the models across the following configurations:

- **Model Sizes**: Small (280K), Medium (2M) and Large (8M).

- **Ordering Strategies**: Random Ordering, Canonical Ordering, and our proposed RC-Rooted Ordering.

- **Sampling Steps**: 10, 20, 50, and 100 Euler steps.

**Impact of Ordering and Model Scale.** Figure 7 visualizes the Top-$k$ accuracies across different model sizes (280K, 2M, 8M) and ordering strategies. The results provide compelling evidence for the efficiency of our positional inductive bias: (1) Bias > Scaling: As observed in the Top-1 accuracy, our Small model (280K) trained with RC-Rooted ordering (Ours) achieves a performance that significantly outperforms the Large model (8M) trained with Random or Canonical ordering ($\approx 70\%$ vs. $\approx 50\%$). This implies that providing correct structural guidance is more effective than increasing the parameter count by over $30\times$; (2) Consistent Superiority: The RC-Rooted strategy (Green and Pink bars) consistently dominates the baselines across all metrics (Top-1 to Top-10).

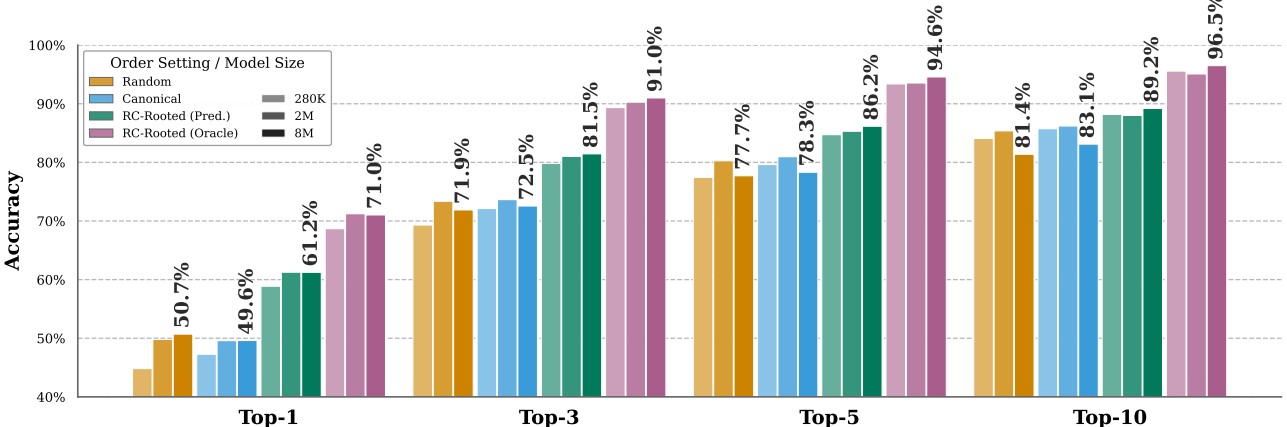

*Figure 7.* **Comprehensive Performance Analysis on USPTO-50k.** Performance comparison across three model sizes (280K, 2M, 8M). The *RC-Rooted* ordering (Green) consistently outperforms *Random* (Yellow) and *Canonical* (Blue) baselines. Notably, the Small (280K) RC-Rooted model outperforms the Large (8M) Random model, highlighting that structure-aware priors are more parameter-efficient than brute-force scaling.

**Chemical Validity and Consistency.** Table 5 details the Round-Trip metrics, assessing whether the generated reactants are chemically valid and can reproduce the target product. Our method demonstrates exceptional chemical plausibility. Under the predicted setting, it achieves a 90.3% Round-Trip Accuracy for Top-1 predictions, surpassing the diffusion baseline RetroBridge (85.1%) and remaining competitive with specialized flow models like RetroProdFlow. Under the Oracle setting, this metric rises to 92.6%, indicating that when conditioned on the correct reaction center, the RetroDiT backbone almost exclusively generates chemically valid and correct reactants. It is worth noting a distinct pattern in the Round-Trip Accuracy: our method achieves superior performance at Top-1 compared to RetroProdFlow (Yadav et al., 2025), while exhibiting slightly lower scores at higher $k$ (e.g., $k = 3, 5, 10$). This behavior underscores the precision-oriented nature of our structure-aware paradigm. The introduction of the *RC-Rooted Inductive Bias* acts as a strong regularization, concentrating the model's probability mass on the most chemically plausible pathway consistent with the predicted reaction center. This leads to a highly 'peaked' distribution where the Top-1 prediction is of exceptionally high fidelity. In contrast,

*Table 5.* **Top-$k$ Round-Trip Coverage and Accuracy on USPTO-50k.** Comparisons with baselines regarding chemical validity and consistency. "–" indicates the metric was not reported in the original paper.

| Model | Round-Trip Coverage (%) | | | | Round-Trip Accuracy (%) | | | |
|---|---|---|---|---|---|---|---|---|
| | $k=1$ | $k=3$ | $k=5$ | $k=10$ | $k=1$ | $k=3$ | $k=5$ | $k=10$ |
| *Template-based* | | | | | | | | |
| GLN (Dai et al., 2019) | 82.5 | 92.0 | 94.0 | – | 82.5 | 71.0 | 66.2 | – |
| LocalRetro (Chen & Jung, 2021) | 82.1 | 92.3 | 94.7 | – | 82.1 | 71.0 | 66.7 | – |
| RetroGFN (Gaiński et al., 2025) | – | – | – | – | 76.7 | 69.1 | 65.5 | 60.8 |
| *Template-free* | | | | | | | | |
| MEGAN (Sacha et al., 2021) | 78.1 | 88.6 | 91.3 | – | 78.1 | 67.3 | 61.7 | – |
| Graph2SMILES (Tu & Coley, 2022) | – | – | – | – | 76.7 | 56.0 | 46.4 | – |
| Ualign (Zeng et al., 2024) | – | – | – | – | 78.6 | 71.8 | 67.1 | – |
| RetroBridge (Igashov et al., 2023) | 85.1 | 95.7 | 97.1 | 97.7 | 85.1 | 73.6 | 67.8 | 56.3 |
| RetroSynFlow-RS (Yadav et al., 2025) | 88.9 | 97.3 | 98.4 | 98.9 | 88.9 | 73.9 | 69.1 | 61.8 |
| RetroProdFlow-RS (Yadav et al., 2025) | 91.4 | 97.6 | 98.7 | 99.3 | 91.4 | **84.1** | **80.1** | **75.1** |
| **Ours (Pred. RC)** | 90.3 | 96.7 | 98.0 | 98.5 | 90.3 | 78.6 | 73.1 | 67.1 |
| **Ours (Orac. RC)** | **92.6** | **98.3** | **99.1** | **99.4** | **92.6** | 79.9 | 74.0 | 68.4 |

baseline methods without such strong structural constraints tend to produce a more diffuse distribution, yielding a diverse set of candidates that sustains accuracy at higher $k$, but lacks the precision of our method at the critical Top-1 rank. For practical retrosynthesis planning, where identifying the single most viable route is often paramount, this trade-off highlights the practical value of our approach.

### F.2. Comparison of Backbones

*Table 6.* **Backbone Comparison: RetroDiT vs. GraphTransformer (GTF).** Evaluated on USPTO-50k with 50 sampling steps using Oracle RC-Rooted ordering.

| Model Size | Backbone | Top-1 | Top-3 | Top-5 | Top-10 |
|---|---|---|---|---|---|
| Small (230K/280K) | GTF | 51.07 | 71.72 | 77.50 | 81.22 |
| | **RetroDiT** | **68.68** | **89.34** | **93.38** | **95.57** |
| Medium (1M/2M) | GTF | 68.56 | 89.15 | 92.79 | 94.72 |
| | **RetroDiT** | **71.23** | **90.27** | **93.54** | **95.08** |
| Large (7M/8M) | GTF | 68.60 | 89.19 | 92.79 | 94.74 |
| | **RetroDiT** | **71.03** | **90.99** | **94.57** | **96.50** |

To validate our architectural choice, we compare the performance of our RetroDiT (which employs RoPE on node embeddings, adaLN, for time conditioning) against a standard Graph Transformer (GTF) baseline (which treats time $t$ as an extra graph level feature added to node embeddings). Both models are evaluated under the Oracle RC-Rooted ordering strategy to isolate the impact of the backbone architecture. Table 6 presents the results at 50 sampling steps.

**Key Observations:**

- **Superior Parameter Efficiency:** RetroDiT demonstrates a significant advantage in parameter efficiency. At the Small scale ($\approx$280K params), RetroDiT outperforms the GTF baseline by a massive margin of over **17%** in Top-1 accuracy (68.68% vs. 51.07%). This indicates that RetroDiT is far more effective at modulating the generative process with time information due to introducing the positional inductive bias when model capacity is limited.

- **Matching Performance at Lower Cost:** Remarkably, the Small RetroDiT (68.68%) achieves performance comparable to the Large GTF (68.60%), which has nearly $30\times$ more parameters. This strongly supports our design choice of using a DiT-based backbone for efficient graph generation.

- **Consistent Scalability:** RetroDiT maintains its performance advantage across all evaluated model scales (Small, Medium, and Large), ensuring superior generative quality without diminishing returns in this regime.

## F.3. Scalability Verification on USPTO-Full

To verify that our findings generalize to larger-scale settings, we evaluate different model sizes and ordering strategies on the USPTO-Full dataset (approximately 1M reactions). Table 7 presents Top-k accuracies across four model scales: Small (280K parameters), Medium (2M), Large (8M), and X-Large (65M), comparing RC-Rooted ordering with predicted and oracle reaction centers against Canonical ordering.

*Table 7.* **Scalability Analysis on USPTO-Full.** Sampling steps set to 50.

| Model Size | Ordering | Setting | Top-$k$ Accuracy (%) | | | |
| --- | --- | --- | --- | --- | --- | --- |
| | | | Top-1 | Top-3 | Top-5 | Top-10 |
| Small (280K) | RC-Rooted | Oracle RC | 50.95 | 67.84 | 72.38 | 76.26 |
| | | Pred. RC | 40.79 | 57.56 | 62.95 | 67.51 |
| | Canonical | – | 36.78 | 55.06 | 61.30 | 66.86 |
| Medium (2M) | RC-Rooted | Oracle RC | 58.24 | 74.19 | 78.21 | 81.45 |
| | | Pred. RC | 46.36 | 63.54 | 68.56 | 72.83 |
| | Canonical | – | 45.98 | 63.51 | 68.73 | 73.12 |
| Large (8M) | RC-Rooted | Oracle RC | 61.80 | 76.76 | 80.35 | 83.15 |
| | | Pred. RC | 49.85 | 66.54 | 71.30 | 75.02 |
| | Canonical | – | 49.31 | 66.11 | 70.73 | 74.44 |
| X-Large (65M) | RC-Rooted | Oracle RC | **63.36** | **77.57** | **80.92** | **83.58** |
| | | Pred. RC | 51.24 | 67.93 | 72.45 | 75.98 |
| | Canonical | – | 50.40 | 67.66 | 72.52 | 76.33 |

**Key Findings:**

**(1) Consistent scaling behavior.** Performance improves steadily as model capacity increases across all settings. From Small to X-Large, oracle RC accuracy improves by 12.41 percentage points (50.95% → 63.36%), while predicted RC and canonical ordering gain 10.45 and 13.62 points respectively. This demonstrates that larger models can better leverage both structural priors and raw learning capacity.

**(2) Oracle RC establishes performance ceiling.** RC-Rooted ordering with oracle reaction centers consistently achieves the best performance at every model scale, reaching 63.36% Top-1 accuracy with the X-Large model. The substantial gap between oracle and predicted RC settings (12.12 points at X-Large) quantifies the upper bound achievable with perfect reaction center identification, validating our framework's potential.

**(3) Diminishing advantage of predicted RC at larger scales.** While RC-Rooted with predicted RCs significantly outperforms Canonical ordering at smaller scales (+4.01 points at Small, +0.38 points at Medium), this advantage narrows and even reverses at larger scales. At X-Large, Canonical ordering achieves slightly better Top-5 and Top-10 accuracies (72.52% vs. 72.45%, 76.33% vs. 75.98%). We attribute this to train-test distribution mismatch: during training, the model learns with ground-truth reaction centers, but at inference, the predicted RCs introduce systematic biases. As model capacity increases, this inconsistency becomes more problematic because larger models may overfit to the ground-truth RC patterns seen during training, making them more sensitive to prediction errors at test time.

**(4) Reinforcing the core thesis: ordering matters.** Importantly, this phenomenon does not undermine our central claim that atom ordering matters. On the contrary, the oracle RC results demonstrate that with accurate reaction centers, RC-Rooted ordering consistently outperforms canonical ordering by substantial margins (12.96 points at X-Large). The challenge lies not in the ordering strategy itself but in the accuracy of the upstream reaction center predictor. This analysis identifies a clear path forward: developing more accurate and calibrated RC predictors that better match the training distribution. Improving RC prediction from its current accuracy to oracle-level performance would unlock the full 12+ point improvement demonstrated in these experiments.

**Implications.** These results validate our framework's scalability while highlighting that reaction center prediction remains the primary bottleneck. The strong oracle performance confirms that our positional inductive bias is effective at scale, and investing in better RC prediction methods represents the most promising direction for future improvement.

## F.4. Sensitivity Analysis: Impact of Reaction Center Accuracy

Complementing the analysis in Section 5.4, we conduct a comprehensive sensitivity analysis to quantify how reaction center (RC) prediction accuracy affects retrosynthesis performance. We evaluate two scenarios: (1) simulated RC accuracy by randomly corrupting ground-truth roots, and (2) actual predicted RCs from our trained predictor. This comparison reveals important insights about the relationship between RC accuracy and generation performance.

**Experimental Setup.** For simulated RC accuracy, we replace ground-truth reaction center atoms with randomly selected atoms with probability $(1 - \text{Acc}\%)$. Critically, in this simulation, we provide the model with the exact number of reaction centers for each molecule, allowing it to focus computational resources appropriately during inference. For the predicted RC setting, we use our trained RC predictor to output top-$k$ candidates without knowing the true number of reaction centers in advance, requiring the model to explore multiple possibilities.

Tables 8 and 9 present detailed performance metrics under various RC accuracy levels for USPTO-50k and USPTO-Full, respectively.

*Table 8.* **Impact of RC Accuracy on USPTO-50k.** Evaluated using RetroDiT-8M with 50 sampling steps. Error bars are computed from 5 independent runs with different random seeds.

| Simulated RC Acc. | Top-1 | Top-3 | Top-5 | Top-10 |
|---|---|---|---|---|
| 10% | 22.47 | 38.26 | 43.99 | 50.74 |
| 20% | 33.07 | 51.41 | 58.13 | 65.39 |
| 30% | 43.64 | 62.82 | 69.43 | 75.56 |
| 40% | 52.34 | 71.33 | 77.38 | 82.90 |
| 50% | 57.52 | 76.51 | 82.24 | 87.60 |
| 60% | 62.27 | 81.04 | 87.17 | 91.48 |
| 70% | 66.52 | 84.46 | 89.32 | 92.96 |
| 80% | 68.04 | 86.67 | 91.58 | 94.57 |
| 85% | 69.55 | 88.55 | 92.92 | 95.32 |
| 90% | 70.44 | 89.21 | 93.38 | 95.57 |
| 95% | 70.97 | 89.80 | 94.05 | 95.99 |
| 100% (Oracle) | 70.30±0.43 | 90.38±0.13 | 93.77±0.18 | 95.52±0.09 |
| 82% (Pred.) | 60.29±0.52 | 80.79±0.31 | 85.70±0.12 | 88.77±0.15 |

*Table 9.* **Impact of RC Accuracy on USPTO-Full.** Evaluated using RetroDiT-8M with 50 sampling steps.

| Simulated RC Acc. | Top-1 | Top-3 | Top-5 | Top-10 |
|---|---|---|---|---|
| 10% | 30.67 | 45.76 | 51.67 | 57.35 |
| 30% | 44.66 | 59.62 | 64.63 | 69.11 |
| 50% | 53.15 | 67.62 | 71.95 | 75.64 |
| 70% | 58.09 | 72.48 | 76.43 | 79.58 |
| 90% | 60.97 | 75.61 | 79.25 | 82.15 |
| 100% (Oracle) | 61.80 | 76.76 | 80.35 | 83.15 |
| 63% (Pred.) | 51.27 | 67.81 | 72.26 | 75.76 |

**Key Observations:**

**(1) Strong correlation between RC accuracy and performance.** Both datasets exhibit a strong positive relationship between RC accuracy and Top-1 accuracy. On USPTO-50k, improving RC accuracy from 10% to 100% yields a 48.64 point gain (22.47% → 71.11%). On USPTO-Full, the gain is 31.13 points (30.67% → 61.80%). This demonstrates that RC prediction is the dominant factor determining overall performance.

**(2) Diminishing returns at high accuracy.** The performance gain per unit improvement in RC accuracy decreases as accuracy increases. On USPTO-50k, improving from 10% to 50% RC accuracy yields +35.05 points (+0.876 per percentage point), while improving from 50% to 100% yields only +13.59 points (+0.272 per percentage point). This suggests that

achieving near-perfect RC prediction may not be necessary; reaching 85-90% accuracy captures most of the potential gains.

**(3) Performance gap between simulated and predicted RCs.** Notably, the predicted RC performance (61.2% on USPTO-50k, 51.3% on USPTO-Full) falls below the simulated accuracy at the same nominal RC accuracy level (68.04% at 80% simulated for USPTO-50k, 53.15% at 50% simulated for USPTO-Full). This discrepancy arises from a fundamental difference in inference setup:

- **Simulated RC**: The model knows the exact number of reaction centers for each molecule. When some roots are randomly corrupted, the model still knows how many RCs exist, allowing it to allocate computational resources efficiently during sampling.

- **Predicted RC**: The RC predictor outputs a fixed number of top-$k$ candidate atoms without knowing the true number of reaction centers. This introduces two sources of error: (a) incorrect RC identification, and (b) computational waste on exploring spurious candidates when the true number of RCs is smaller than $k$, or missing valid RCs when the true number exceeds $k$. This mismatch between predicted candidate set size and true RC count reduces efficiency and performance.

**(4) Dataset-specific sensitivity.** USPTO-50k shows higher absolute performance and larger performance gaps between different RC accuracy levels compared to USPTO-Full. This suggests that USPTO-50k reactions may have more predictable patterns that benefit more strongly from accurate RC identification, while USPTO-Full's greater diversity makes it inherently more challenging regardless of RC accuracy.

This analysis confirms that RC prediction accuracy is the primary bottleneck and quantifies the potential gains from improved RC prediction. However, it also reveals that simply improving RC classification accuracy is insufficient; the predictor must also reliably estimate the number of reaction centers to avoid computational waste during generation. Future work should focus on developing RC predictors that output calibrated confidence scores and variable-size candidate sets matched to each molecule's true complexity, rather than fixed top-$k$ predictions.

### F.5. Performance Breakdown by Reaction Type

**Performance Analysis by Reaction Class.** To understand how our method performs across different types of chemical transformations, we analyze accuracy by the 10 standard reaction classes in USPTO-50k. Figure 8 shows the class distribution across train, validation, and test splits, while Figures 9a and 9b present Top-k accuracies with predicted and oracle reaction centers, respectively.

**Key Observations:**

- **Class imbalance effect**: Performance correlates with class frequency. Classes 1–2 (most frequent, >25% each) achieve strong Top-1 accuracies of 65.4–72.6% with predicted RCs and 73.5–79.4% with oracle RCs. In contrast, underrepresented classes like Class 9 (<5%) show the poorest performance (40.0% predicted, 53.7% oracle).

- **Exceptional performance on Class 5**: Despite being relatively rare ($\approx$5%), Class 5 achieves remarkable accuracy (86.5% predicted, 89.2% oracle), reaching 100% Top-3 accuracy with oracle RCs. This suggests these reactions have highly predictable patterns that our ordering strategy captures effectively.

- **Most challenging classes**: Classes 3, 7, and 9 show the largest performance gaps and lowest absolute accuracies. With predicted RCs, Class 7 achieves only 38.7% Top-1 accuracy, and Class 9 reaches 40.0%. These classes likely involve complex structural rearrangements that are difficult to predict.

- **Oracle vs. Predicted gap**: The performance difference between oracle and predicted RCs varies by class. Classes 3, 6, and 7 show the largest gaps, indicating that accurate RC prediction is particularly critical for these reaction types. Class 10, despite small sample size, maintains strong performance (80% across both settings), suggesting robust learned patterns.

- **Top-k performance**: All classes benefit substantially from Top-k predictions. Even the most challenging classes (3, 7, 9) achieve >75% Top-10 accuracy with predicted RCs and >87% with oracle RCs, demonstrating that correct reactants often appear in the candidate set even when not ranked first.

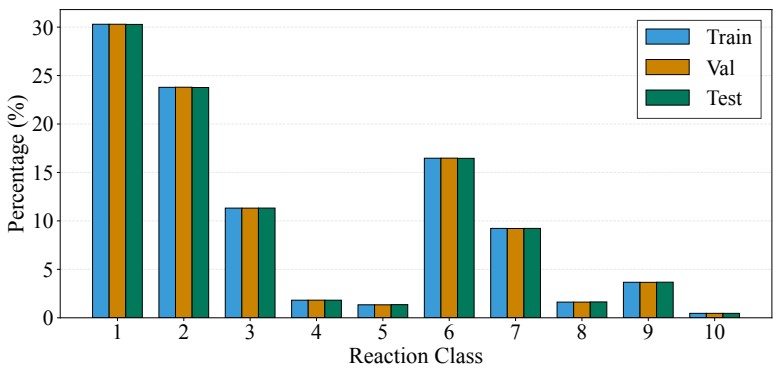

*Figure 8.* **Reaction Class Distribution.** Classes show consistent distributions across splits.

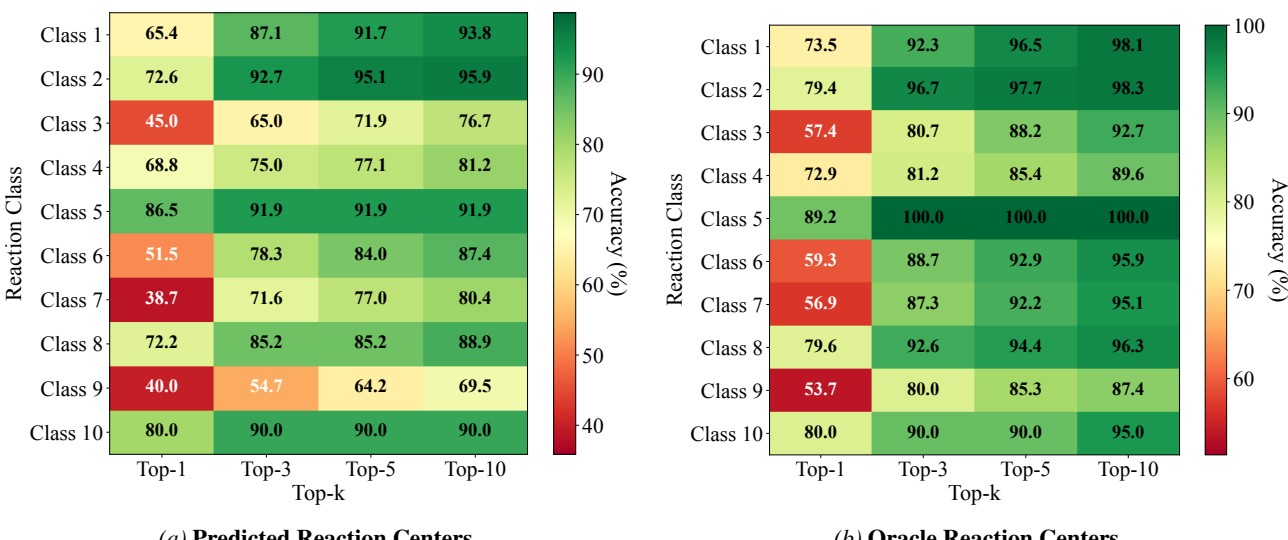

| *(a)* **Predicted Reaction Centers** | *(b)* **Oracle Reaction Centers** |

*Figure 9.* **Top-k Accuracy Heatmaps by Reaction Class.** The comparison reveals that RC prediction accuracy significantly impacts certain classes more than others.

This analysis reveals that while our positional inductive bias is effective across diverse reaction types, performance is influenced by both class frequency (data availability) and intrinsic chemical complexity. The strong performance on well-represented classes and the consistent benefit of oracle RCs underscore that improving RC prediction remains the key to unlocking better performance, particularly for underrepresented and structurally complex reaction classes.

## G. Training Dynamics and Efficiency Analysis

This section provides a detailed analysis of the training dynamics and computational efficiency of our framework.

### G.1. Training Dynamics

To demonstrate how the positional inductive bias reduces learning difficulty, we compare the training dynamics of the RetroDiT-8M model under three ordering strategies: RC-Rooted (Ours), Canonical, and Random. Figure 10 displays the training curves. Our proposed RC-Rooted method achieves lower training loss (a-b) and validation loss (d-f) compared to Canonical and Random Order baselines, while maintaining superior Top-$k$ accuracy (g-i). Notably, the validation curves of RC-Rooted continue to improve throughout training without signs of overfitting, whereas canonical order exhibits clear overfitting behavior after epoch 150 with rising validation loss despite decreasing training loss.

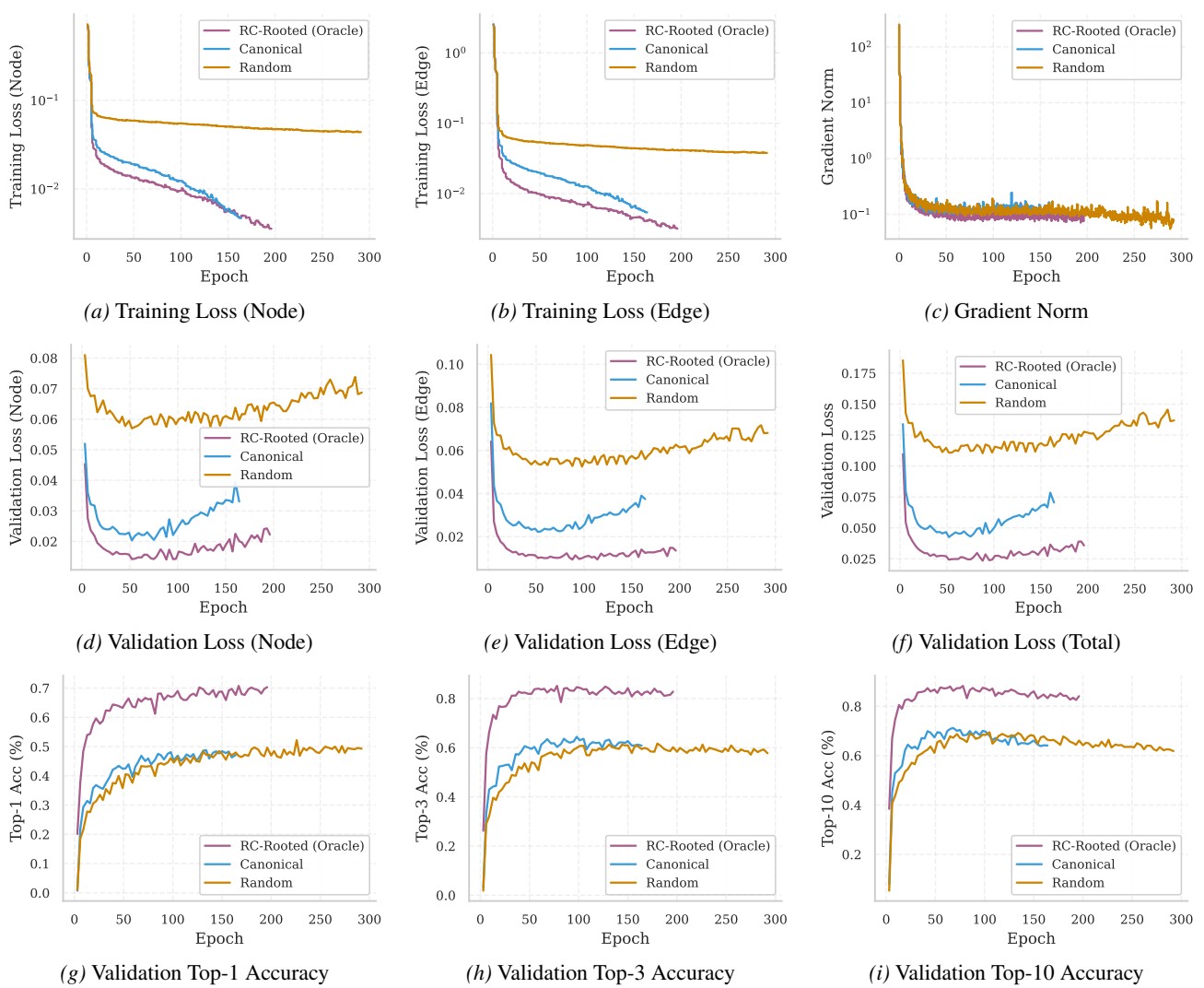

*Figure 10.* **Training dynamics comparison on USPTO-50k**.

## G.2. Wall-Clock Training Time

We compare the computational cost of our method against RetroBridge (Igashov et al., 2023), the primary generative baseline. The comparison is based on the time required to reach model convergence. As shown in Table 10, our framework achieves a **6×** **reduction in total GPU-hours** compared to RetroBridge. While RetroBridge requires approximately 3 days (72 hours) on a single GPU, our method converges in just 1.5 hours using 8 GPUs (equivalent to 12 GPU-hours).

*Table 10.* **Training Cost Comparison.** Time to convergence on USPTO-50k.

| Model | Hardware | Wall-Clock Time | Total GPU-Hours | Speedup |
| --- | --- | --- | --- | --- |
| RetroBridge (Igashov et al., 2023) | $1 \times$ GPU | $\sim 72.0$ h | 72.0 | $1.0\times$ |
| **Ours** | $8 \times$ GPU | $\sim$ **1.5 h** | **12.0** | **6.0×** |

## G.3. Sampling Efficiency

In this part, we analyze the inference efficiency by evaluating the generation performance across different numbers of Euler sampling steps (Function Evaluation Evolutions, NFEs). Figure 11 plots the Top-1 Accuracy of RetroDiT against the number of sampling steps ($N \in \{10, 20, 50, 100\}$).

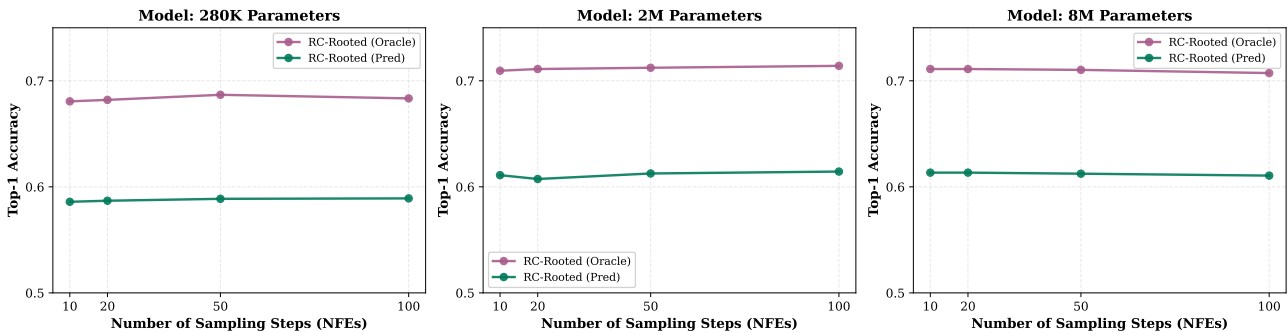

*Figure 11.* **Sampling Efficiency.** Top-1 Accuracy on USPTO-50k vs. Number of Sampling Steps. Performance saturates at around 20 steps, demonstrating high inference efficiency.

**Key Observation.** Our method achieves near-optimal performance with as few as 20 steps. Increasing the steps to 50 or 100 yields negligible marginal gains. In contrast, standard discrete diffusion models typically require 500-1000 steps to generate high-quality graphs. This efficiency is a direct benefit of the linear interpolation path used in DeFoG (Qin et al., 2024). By constructing a straight probability path from the product to the reactant under the dat-aligned fashion, the trajectory is much simpler to integrate than the curvature of diffusion paths, allowing for significantly larger step sizes $\Delta t$ without accumulating discretization errors.

## H. Qualitative Analysis and Visualization

This section provides qualitative insights into the internal mechanisms and generative capabilities of our framework. We visualize attention maps to validate our structural hypothesis, showcase diverse generation examples, analyze failure modes, and illustrate the dynamic evolution of the discrete flow.

### H.1. Attention Mechanism Analysis

To empirically validate the Head-to-Tail Interaction hypothesis proposed in Section 4.2, we analyze and visualize the self-attention patterns learned by the RetroDiT-8M backbone under different SMILES ordering strategies. Specifically, we compare three settings: the proposed *reaction-centered (RC)* ordering, *canonical* order SMILES, and *random* order SMILES without positional encoding.

Figure 12 presents representative attention heatmaps across layers and heads for the three settings. While attention weights should not be interpreted as direct explanations, their statistical structure provides a useful probe for understanding how different input orderings induce distinct inductive biases in the Transformer backbone.

**Qualitative observations.** The random ordering results in highly fragmented and noisy attention patterns, with little consistency across layers. In contrast, canonical SMILES induces strongly diagonal and highly repetitive attention across heads, reflecting a dominant reliance on sequential adjacency. Notably, the RC ordering exhibits neither extreme: attention patterns gradually evolve across layers, and different heads attend to complementary regions of the sequence, including both product atoms and reactant-only dummy nodes. This behavior is consistent with the intended head-to-tail interaction mechanism, where reactant-only atoms are selectively activated based on their relevance to the reaction core.

**Quantitative attention statistics.** To further support these observations, we compute a set of standard attention metrics commonly used in prior interpretability and representation analysis studies, including attention entropy (Zhai et al., 2023), head diversity (Li et al., 2018), column dominance (Xiao et al., 2023), layer-wise similarity (Kornblith et al., 2019), and diagonal dominance (Shi et al., 2021). These metrics are evaluated at both the first and final diffusion steps; the results are summarized in Table 11.

**Quantitative interpretation.** The attention statistics in Table 11 reveal several key differences that support our hypothesis:

1. **Attention Entropy:** The RC ordering exhibits a distinctive pattern: entropy increases from the first to final diffusion

*Table 11.* **Attention statistics at different diffusion steps.** Quantitative attention metrics computed from the self-attention weights of RetroDiT-8M under three SMILES ordering strategies. Results are reported at the first and final diffusion steps.

| Metric | Random | | Canonical | | Reaction-Centered | |
|---|---|---|---|---|---|---|
| | **First** | **Final** | **First** | **Final** | **First** | **Final** |
| Attention Entropy | 1.8581 | 1.7153 | 1.6835 | 1.7090 | 1.7792 | 1.8576 |
| Head Diversity | 0.0265 | 0.0280 | 0.0293 | 0.0288 | 0.0247 | 0.0239 |
| Column Dominance | 8.6272 | 11.817 | 7.5721 | 9.3836 | 8.2850 | 8.9847 |
| Layer Similarity | 0.2090 | 0.2117 | 0.1642 | 0.1861 | 0.1883 | 0.2137 |
| Diagonal Dominance | 0.2662 | 0.1956 | 0.2225 | 0.1882 | 0.2226 | 0.1814 |

step, whereas the random ordering shows the opposite trend. This suggests that under RC ordering, the model progressively broadens its attention scope as generation proceeds—consistent with the intuition that early steps focus on the reaction center while later steps distribute attention toward completing the leaving groups at the sequence tail.

2. **Head Diversity:** The RC ordering yields the lowest head diversity scores, indicating that different attention heads converge to more similar patterns. We hypothesize that this reflects the strong structural prior imposed by RC ordering: when the reaction center is consistently placed at the sequence head, the model receives clear positional guidance, reducing the need for different heads to "explore" diverse attention strategies. In contrast, the random and canonical orderings exhibit higher head diversity, potentially because the model must hedge across multiple attention patterns to compensate for the lack of consistent structural cues. Notably, despite lower head diversity, RC ordering achieves superior generation performance, suggesting that aligned attention heads attending to semantically meaningful positions are more effective than diverse heads attending to arbitrary regions.

3. **Column Dominance:** The random ordering exhibits a dramatic increase in column dominance, suggesting that certain tokens become disproportionately dominant sinks for attention, a pathological pattern often associated with poor representation learning. In contrast, RC ordering maintains relatively stable column dominance, indicating more balanced information flow across the sequence.

4. **Layer Similarity**: The RC ordering exhibits the largest increase in layer similarity from first to final diffusion step, achieving the highest final value among all orderings. This suggests that as generation proceeds, attention patterns across layers progressively converge toward the same semantically relevant positions, namely the reaction center and associated leaving groups. In contrast, canonical ordering maintains persistently low layer similarity, indicating that layers operate more independently without a unified focus due to the lack of explicit structural guidance.

5. **Diagonal Dominance:** All three orderings show decreasing diagonal dominance from first to final step, reflecting a shift from local to global attention as the diffusion process converges. Notably, RC ordering achieves the lowest final diagonal dominance, confirming that the model learns to attend beyond immediate sequential neighbors, a prerequisite for effective head-to-tail interaction between reaction center atoms and distant dummy nodes.

Collectively, these metrics provide quantitative evidence that RC ordering induces more structured, diverse, and semantically meaningful attention patterns compared to both random and canonical alternatives.

### H.2. Generative Trajectory Visualization

Finally, we visualize the discrete flow matching process. Figure 13 displays snapshots of a graph evolving from $t = 0$ to $t = 1$ via the Euler sampling process.

**Evolution**:

- At $t = 0$, the graph is the Product, and Dummy Nodes are "Empty".

- As $t$ increases, the probability mass shifts linearly. We observe bonds at the reaction center gradually breaking (edge features changing) and Dummy Nodes progressively taking on the identity of the reactant atoms.

- At $t = 1$, the structure stabilizes into the final Reactants.

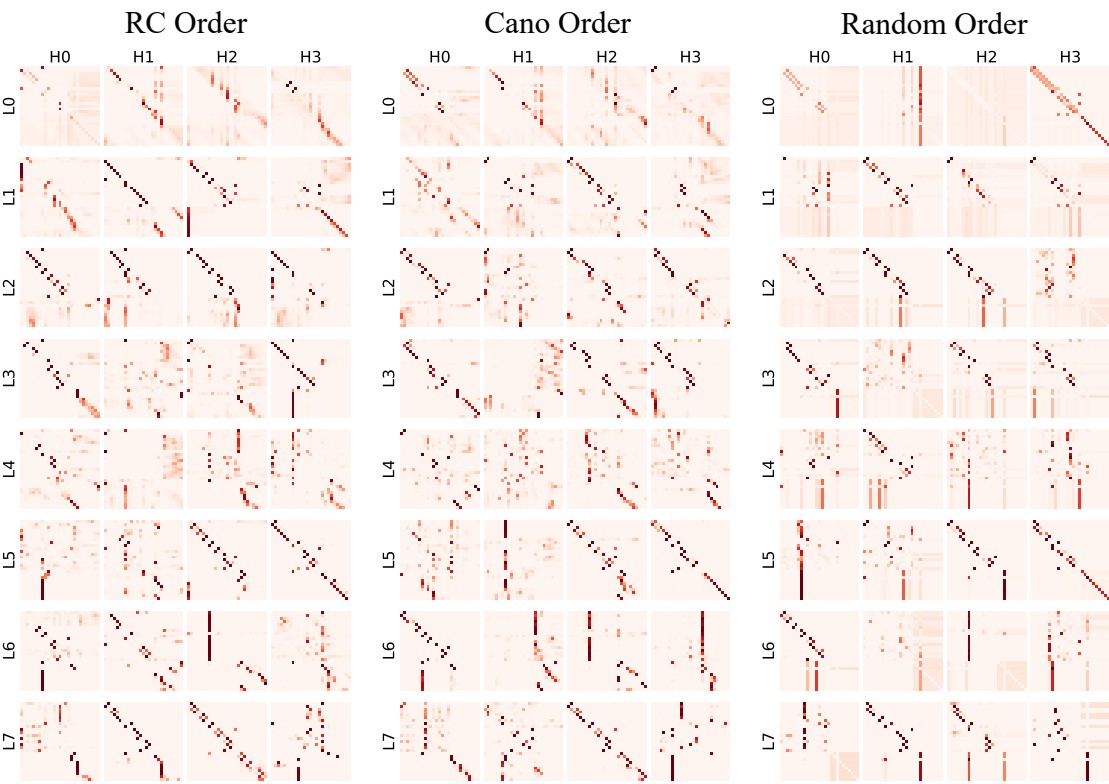

*Figure 12.* **Visualization of Head-to-Tail Attention.** Heatmaps show strong attention from Dummy Nodes (Tail) to the Reaction Center (Head), confirming that RoPE facilitates the broadcasting of structural constraints to leaving groups.

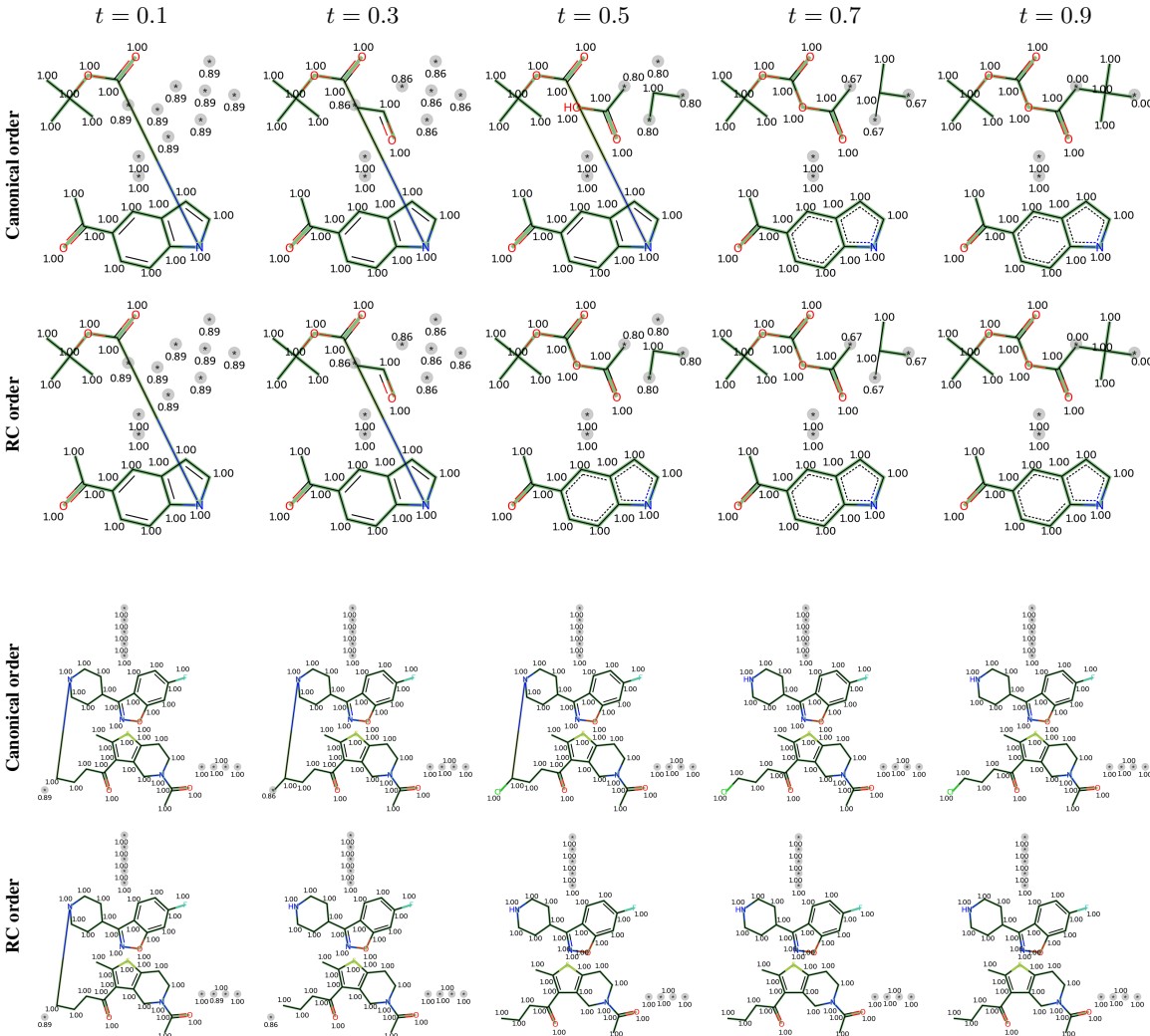

*Figure 13.* **Generative Trajectory.** Snapshots of the graph generation process from $t = 0$ (Product) to $t = 1$ (Reactants), showing the progressive filling of dummy nodes and bond modification.

