# OpenReview forum: "Order Matters in Retrosynthesis: Structure-aware Generation via Reaction-Center-Guided Discrete Flow Matching"
_ICML.cc/2026/Conference — ICML 2026 regular_

### Official Review · Reviewer_2ANh · 2026-02-22

**Soundness:** 2
**Presentation:** 2
**Significance:** 3
**Originality:** 2
**Overall Recommendation:** 4
**Confidence:** 4

**Summary:**

This manuscript describes RetroDiT, which approaches the chemistry task of one-step retrosynthesis in a two-stage manner: first, predicting reacting atoms with an R-GCN; second, reordering atoms and using a graph transformer with RoPE to break permutation equivariance. The generator backbone is a discrete flow matching model. Quantitative results are reported on USPTO-50k and USPTO-Full, with additional analyses related to the role of the reaction center prediction model in upper bounding accuracy.

**Compliance With Llm Reviewing Policy:**

Affirmed.

**Final Justification:**

The clarifications the authors have made in their rebuttal have helped. The corrections to presenting prior work fairly are strong. I still believe that there are weaknesses with regards to novelty, as previously stated. The empirical results look strong but I retain skepticism about their validity without being able to reproduce them fully

**Key Questions For Authors:**

1.	There is not a major conceptual distinction between the use of a graph transformer with RoPE to center atom numbering on reacting atoms and using a reacting atom-aligned SMILES string with a SMILES transformer. Reacting atoms can also, for a permutation-equivariant graph neural network or graph transformer, be tagged through atom-level features. Were either of the plausible alternatives to RoPE mentioned above evaluated (using atom features only for the reacting atoms or using RC-aligned SMILES with a SMILES Transformer?
2.	Can the authors separate the RC-known from the other evaluations, as this represents a distinct and not-so-realistic task?

**Limitations:**

yes

**Strengths And Weaknesses:**

-	Retrosynthesis is an important task in chemistry. Although benchmarks are mostly saturated, in principle any improvement could translate well to improved performance in prospective settings.
-	Prior work has already decomposed the one-step retrosynthesis task into two stages of (1) reaction center identification and (2) reactant generation without a fixed vocabulary. For example, RetroXpert and RetroPrime do so through the generation of intermediate synthons after tagging predicted reacting atoms. 10.1021/acscentsci.3c00372 does this with a Transformer that simply adds tokens to the reacting atoms in the first stage, and then generates reactants in the second. Several prior studies thus feature this notion of “upgradability” and diminish the conceptual novelty of this work.
-	The “Oracle RC” setting bears little resemblance to a realistic chemistry task and is inappropriate to include in a comparison table to other methods that do not prespecify the reaction center. Certain prior studies have also evaluated this setting; if Oracle RC results are to be included, comparisons must be drawn to these as well. (To my knowledge, most of these exhibit top-1 scores of <70%, but Retro-MTGR (10.1038/s41467-025-56062-y) might be an exception.)
-	Minor: Semi-template based methods are described as relying on templates or predefined rules for completing synthons; this is not true of all of the cited papers and methods commonly used under this grouping, to my knowledge.

---

### Official Review · Reviewer_QHU1 · 2026-03-12

**Soundness:** 3
**Presentation:** 2
**Significance:** 3
**Originality:** 2
**Overall Recommendation:** 3
**Confidence:** 4

**Summary:**

This paper studies template-free retrosynthesis. The method puts r-smiles ordered RoPE embedding on the product's mol graph, which is then processed through a flow-matching discrete graph DiT to generate the reactant. The experimental results are comparable to SOTA methods on standard benchmarks.

**Compliance With Llm Reviewing Policy:**

Affirmed.

**Final Justification:**

see the official comment

**Key Questions For Authors:**

1. The rsmiles ordering can partially reflect distance to the reaction center. But molecular structure mainly lives in 3D Euclidean space, while rsmiles is a 1D sequence. This difference may introduce bias. Did the authors consider this mismatch and have evidence that this prior does not hurt generalization?

2. The paper uses an extra RGCN to predict reaction centers. What's the accuracy of this RC predictor? Did the authors try stronger predictors, or a unified multitask backbone shared by reaction-center prediction and retrosynthesis?

**Limitations:**

See strengths and weaknesses.

**Strengths And Weaknesses:**

It's interesting that introducing string-order embeddings to the mol graph for structural modeling. The pipeline design of RGCN->rsmiles->DiT is sound and concise.

However, the novelty is still limited as core components, including reaction-center-aware R-SMILES, incoporation of semi-template knowledge (RetroEdit), and flow-matching discrete graph DiT (RetroSynFlow), have all been established in prior literature.

The manuscript is mostly clear, but it spends too much space re-introducing known components like rsmiles, flow-matching and discrete diffusion. The paper should focus more on what is truly new here (e.g. the integration of string and graph mol modeling), and put those existing concepts in citation or appendix.

The benchmark choice is standard, but lack evaluation on USPTO-MIT (and some baselines on USPTO-FULL). The gain over existing methods is marginal (61.2 vs 60.8). The authors compare their model with oracle reaction centers against RSGPT in Section 5.2, which is an unfair comparison. I also recommend that the authors report the mean/std result of the proposed method across >=3 trarining random seeds (for both dataset iterator and param initializer), as in RetroSynFlow.

---

### Official Review · Reviewer_mFRE · 2026-03-13

**Soundness:** 2
**Presentation:** 3
**Significance:** 2
**Originality:** 3
**Overall Recommendation:** 3
**Confidence:** 4

**Summary:**

The authors propose RetroDiT, a framework that treats retrosynthesis as a structure-aware generation process. Their key contribution is the RC-rooted atom ordering, which places reaction center atoms at the head of the sequence to provide a positional inductive bias, combined with Rotary Position Embeddings (RoPE) and Discrete Flow Matching (DFM).

**Compliance With Llm Reviewing Policy:**

Affirmed.

**Final Justification:**

Despite additional experiments on scalability and class-known settings, core concerns remain, regarding the performance bottleneck and error propagation caused by the upstream predictor, as well as the questionable robustness of the positional bias given the model's unexpected tolerance for incorrect RCs. Furthermore, the reliance on the USPTO benchmark alone leaves the model’s generalizability across diverse chemical spaces unproven.

**Key Questions For Authors:**

See Weaknesses.

**Limitations:**

The authors have not discussed limitations. The paper should acknowledge the method's heavy reliance on reaction center accuracy and note that experiments are limited to relatively simple USPTO datasets, leaving generalizability to more complex reactions unclear.

**Strengths And Weaknesses:**

Strengths:
1. The paper is well-written and easy to follow.
2. The proposed reaction-center-first atom ordering is conceptually simple and aligns with chemical intuition that reaction centers determine the transformation.
3. The reported results on standard benchmarks show improvements over several baseline methods, and the ablation studies provide some evidence that atom ordering contributes to performance gains.

Weaknesses:
1. The method introduces a fixed number of dummy nodes to accommodate leaving groups in the framework. This design may introduce limitations when reactions involve leaving groups with larger or more complex structures than the predefined capacity.
2. The inference procedure requires predicting the top-k reaction centers (RCs). Does it increase the actual wall-clock inference time compared to fully end-to-end Transformer-based retrosynthesis models?
3. The reported results show that when the RC prediction accuracy is around 35-40%, the final retrosynthesis top-1 accuracy reaches around 50%. This raises some questions about how the model recovers correct predictions when the predicted reaction center is incorrect.
4. The evaluation focuses primarily on the class-unknown setting. However, reaction centers are often closely related to reaction classes, and therefore the class-known setting could provide additional insights into how well the model captures reaction patterns. Does predicting the reaction center implicitly leak reaction class information, potentially leading to an unfair comparison with the baselines?

---

### Official Review · Reviewer_uBfj · 2026-03-13

**Soundness:** 3
**Presentation:** 3
**Significance:** 3
**Originality:** 3
**Overall Recommendation:** 5
**Confidence:** 4

**Summary:**

In this paper, the authors proposed RetroDiT, a graph-neural network-based retrosynthesis prediction model with rotary positional encoding. In this work, the authors proposed the retrosynthesis prediction method, RetroDiT, using discrete flow matching and ground truth reaction center obtained from atom mapping. During training, the discrete flow matching model was trained to predict the probability of intermediate step, G_t. During training, SMILEs strings were re-ordered so that the reaction center atom is placed at the first. During the test time, the retrosynthesis route was obtained using continuous time Monte Carlo simulation using the trained RetroDiT model.
The benchmark results show that the proposed method achieved high top-k prediction accuracy compared to many existing baseline methods.

**Compliance With Llm Reviewing Policy:**

Affirmed.

**Final Justification:**

The authors addressed my concerns and I keep my initial verdict.

**Key Questions For Authors:**

1) How were hyperparameters optimized?
2) How were the hyperprameters of RoPE set?
3) Did the method result in all valid SMILES? How high was the validity of generated SMILES strings?

**Limitations:**

Yes

**Strengths And Weaknesses:**

Strengths:
1) This work presents a novel framework to perform retrosynthesis prediction by combining discrete flow matching with reaction center prediction. Decoupling training and inference phases is also an effective way to train an optimal model.

2) The performance of RetroDiT is comparable or superior to existing template-free and template-based methods.

3) Placing a reaction site at the first position and using RoPE to employ this information is an effective strategy for retrosynthesis prediction and the authors showed the effect of RoPE by performing proper ablation studies.

Weakness:
1) Although the authors performed extensive comparison with many baseline methods, template-based and semi-template-based methods used for comparison have been suggested several years ago. More recent template-based models should be used for more rigorious benchmarking.

---

### Decision · Program_Chairs · 2026-04-30

**Decision:**

Accept (regular)

**Comment:**

This paper proposes RetroDiT, a structure-aware template-free retrosynthesis method that encodes reaction center information as a positional inductive bias through atom ordering. The method places reaction center atoms at the head of the node sequence, uses RoPE in a graph transformer, and employs discrete flow matching for generation. Results on USPTO-50k and USPTO-Full show competitive performance with significantly fewer parameters and faster training than prior methods.

This is a borderline paper whose review process was complicated by the authors submitting their rebuttal through the AC-only channel rather than the official system. I forwarded the rebuttal to all reviewers and have read both the reviews and rebuttals carefully to ensure a fair assessment.

The reviewers raised four main concerns: limited novelty since individual components exist in prior work; marginal Top-1 improvement over the closest baseline; heavy dependence on the upstream RC predictor; and misleading inclusion of oracle RC results alongside methods that do not use oracle information. The authors provided rebuttals addressing all concerns, but Reviewers mFRE and QHU1 were not persuaded, leading to a borderline score distribution (3/3/4/5).

From my own reading, I believe the idea is clean and well-motivated, and there are real improvements in Top-3/5/10 accuracies and training efficiency — the latter being an underappreciated but practically significant contribution. The most critical weakness is that Top-1 improvement over RetroSynFlow is modest at best, but the consistent gains at higher k and the strong efficiency story compensate for this.

I recommend borderline acceptance.